# TAF15 amyloid filaments in frontotemporal lobar degeneration

Stephan Tetter[1], Diana Arseni[1], Alexey G. Murzin[1], Yazead Buhidma[2], Sew Y. Peak-Chew[1], Holly J. Garringer[3], Kathy L. Newell[3], Ruben Vidal[3], Liana G. Apostolova[4], Tammaryn Lashley[2,5], Bernardino Ghetti[3] & Benjamin Ryskeldi-Falcon[1✉]

Frontotemporal lobar degeneration (FTLD) causes frontotemporal dementia (FTD), the most common form of dementia after Alzheimer's disease, and is often also associated with motor disorders[1]. The pathological hallmarks of FTLD are neuronal inclusions of specific, abnormally assembled proteins[2]. In the majority of cases the inclusions contain amyloid filament assemblies of TAR DNA-binding protein 43 (TDP-43) or tau, with distinct filament structures characterizing different FTLD subtypes[3,4]. The presence of amyloid filaments and their identities and structures in the remaining approximately 10% of FTLD cases are unknown but are widely believed to be composed of the protein fused in sarcoma (FUS, also known as translocated in liposarcoma). As such, these cases are commonly referred to as FTLD–FUS. Here we used cryogenic electron microscopy (cryo-EM) to determine the structures of amyloid filaments extracted from the prefrontal and temporal cortices of four individuals with FTLD–FUS. Surprisingly, we found abundant amyloid filaments of the FUS homologue TATA-binding protein-associated factor 15 (TAF15, also known as TATA-binding protein-associated factor 2N) rather than of FUS itself. The filament fold is formed from residues 7–99 in the low-complexity domain (LCD) of TAF15 and was identical between individuals. Furthermore, we found TAF15 filaments with the same fold in the motor cortex and brainstem of two of the individuals, both showing upper and lower motor neuron pathology. The formation of TAF15 amyloid filaments with a characteristic fold in FTLD establishes TAF15 proteinopathy in neurodegenerative disease. The structure of TAF15 amyloid filaments provides a basis for the development of model systems of neurodegenerative disease, as well as for the design of diagnostic and therapeutic tools targeting TAF15 proteinopathy.

Neuronal inclusions containing abnormally assembled TDP-43 or tau characterize approximately 50% and 40% of FTLD cases, respectively[2]. The assemblies have amyloid structure[3–5]. Amyloids are filamentous protein assemblies stabilized by intermolecular β-sheets along the filament axis. Although the proteins are wild type in most cases of disease, rare mutations in the genes encoding TDP-43 and tau that give rise to amyloid assembly and FTLD demonstrate a causal link[6–10]. Furthermore, distinct amyloid filament folds of TDP-43 and tau define different subtypes of FTLD[3–5], which are associated with various behavioural and language variants of FTD, as well as with motor disorders[11].

By contrast, the presence, identities and structures of amyloid filaments within the neuronal inclusions of the remaining approximately 10% of FTLD cases are unknown. The inclusions were initially found to be immunoreactive for FUS, resulting in these cases being commonly referred to as FTLD–FUS[12–15]. The search for FUS was motivated by the discovery that rare mutations in *FUS* can cause the motor disorder amyotrophic lateral sclerosis (ALS) in the absence of FTLD[16,17].

Furthermore, recombinant fragments of the FUS LCD can assemble into amyloid filaments in vitro[18–20]. However, to date, mutations in *FUS* associated with FTLD have not been found[21,22] and amyloid filaments of FUS have not been identified in patient brains.

It was subsequently shown that the inclusions of FTLD–FUS are also immunoreactive against TAF15 and transportin 1 (also known as importin β-2 and karyopherin β-2)[23–27]. For some of the cases in these studies, a subset of inclusions was also immunoreactive against Ewing's sarcoma (EWS). FUS, EWS and TAF15 are homologous RNA-binding proteins, collectively known as the FET proteins[28]. Owing to FET protein immunoreactivity, FTLD–FUS has also been referred to as FTLD–FET[23], a more comprehensive term that we, therefore, use from here on. Evidence suggests that, like FUS, the LCDs of TAF15 and EWS can also assemble into filaments in vitro[29–32].

In healthy cells, FET proteins are mainly localized in the nucleus and undergo nucleocytoplasmic shuttling[28]. FET proteins have roles in transcription and in the splicing, processing and transport of RNA.

[1]MRC Laboratory of Molecular Biology, Cambridge, UK. [2]Department of Neurodegenerative Diseases, UCL Queen Square Institute of Neurology, London, UK. [3]Department of Pathology and Laboratory Medicine, Indiana University School of Medicine, Indianapolis, IN, USA. [4]Department of Neurology, Indiana University School of Medicine, Indianapolis, IN, USA. [5]The Queen Square Brain Bank for Neurological Disorders, Department of Clinical and Movement Neuroscience, UCL Queen Square Institute of Neurology, London, UK. ✉e-mail: bfalcon@mrc-lmb.cam.ac.uk

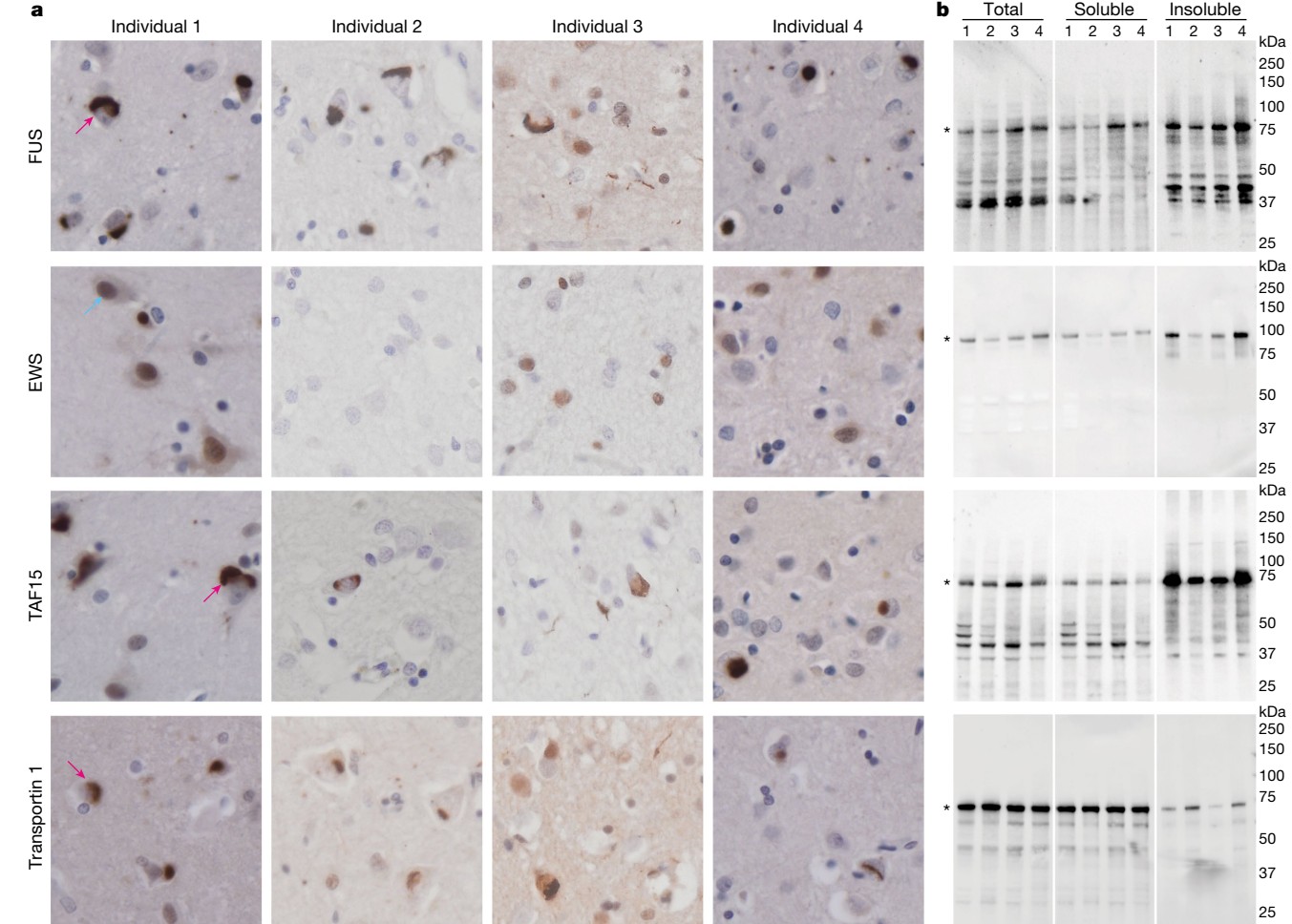

**Fig. 1 | FET proteins and transportin 1 in FTLD–FET. a**, FUS, EWS, TAF15 and transportin 1 immunoreactivity (brown) in the prefrontal cortex of individials 1–4 with FTLD–FET. Sections were counterstained with haematoxylin (blue). Scale bar, 50 μm. Neuronal cytoplasmic inclusions were immunoreactive for FUS, TAF15 and transportin 1 (examples indicated by magenta arrows for individual 1). Antibodies against EWS showed diffuse labelling of nuclei only (example indicated by cyan arrow for individual 1). **b**, Immunoblots of the total homogenate, sarkosyl-soluble fraction and sarkosyl-insoluble fraction of frontotemporal cortex grey matter from individuals 1–4 with FTLD–FET with antibodies against FUS, EWS, TAF15 and transportin 1. Asterisks indicate bands corresponding to full-length proteins. Bands of lower molecular weight probably correspond to protease cleavage products. For uncropped images of immunoblots see Supplementary Fig. 1. **a**,**b**, Results are representative of $n \geq 3$ technical replicates per individual.

Their N-terminal LCDs are enriched in glycine, tyrosine, glutamine and serine residues. They also contain a mid-region RNA recognition motif flanked by arginine–glycine–glycine (RGG) motif-rich segments, a zinc finger domain and a C-terminal nuclear localization signal (NLS). Transportin 1 binds to the NLS of FET proteins to mediate their nuclear import[33].

To understand neurodegeneration at a molecular level and to provide a basis for diagnostic and therapeutic strategies, a structural understanding of pathological protein assembly is essential[34,35]. Here we investigated the presence, identities and structures of amyloid filaments in the brains of individuals with FTLD–FET.

## Amyloid filaments in FTLD–FET

We analysed tissue from the prefrontal and temporal cortices of four individuals with FTLD–FET (Extended Data Table 1). Immunohistochemistry using antibodies against FUS, TAF15 and transportin 1 confirmed the presence of abundant neuronal cytoplasmic inclusions, and occasional neuronal intranuclear inclusions and glial cytoplasmic inclusions (Fig. 1a), as previously reported[23–27]. We did not detect inclusions using an antibody against EWS (Fig. 1a), consistent with previous reports of scarce or absent EWS inclusion immunoreactivity in FTLD–FET[23,26,27].

We extracted insoluble material from tissues using differential centrifugation in the presence of the detergent *N*-lauroyl-sarcosine (sarkosyl). This method enriches for stable protein assemblies, including amyloid filaments from human brain[3,5,36–38]. Negative-stain electron microscopy of samples from individual 1 showed amyloid filaments in the insoluble fraction whereas none were observed in the soluble fraction (Extended Data Fig. 1a). Immunoblotting showed the presence of all three FET proteins and transportin 1 in the insoluble fraction (Fig. 1b). Previous studies have also observed these proteins in detergent-insoluble fractions of human brain by immunoblotting, including from neurologically normal individuals[12,15,23–25]. These results suggest that FET proteins and transportin 1 form detergent-stable assemblies in human brain but are not sufficient to determine whether they form amyloid filaments.

We used cryo-EM to image amyloid filaments in the insoluble extracts from each individual. The majority of filaments were approximately 8 nm in width (Fig. 2a and Extended Data Fig. 1b). A minority of filaments from individuals 2–4 could be identified as those of transmembrane protein 106B (TMEM106B) based on their distinct width of either 12 nm (single protofilament) or 26 nm (double protofilament), smooth surface, striated appearance and blunt filament ends[38–40] (Extended Data Fig. 1b). We determined high-resolution cryo-EM structures of these

**a**

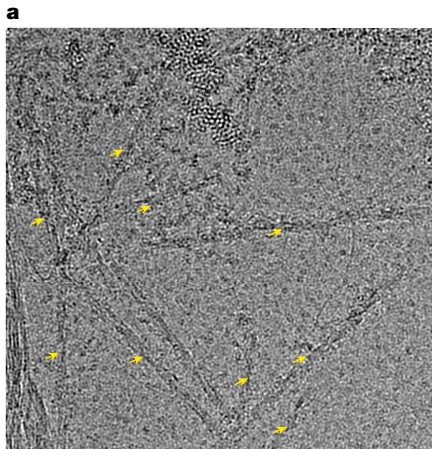

**b**

Individual 1    Individual 2

2.0 Å    2.1 Å

Individual 3    Individual 4

2.6 Å    2.7 Å

**Fig. 2 | Cryo-EM characterization of amyloid filaments from individuals with FTLD–FET. a**, Representative cryo-EM micrograph of the sarkosyl-insoluble fraction of frontotemporal cortex grey matter from individual 1 with FTLD–FET. Abundant amyloid filaments are indicated by arrows. Scale bar, 50 nm. Results are representative of $n \geq 3$ technical replicates per individual.

Additional micrographs for all four individuals are shown in Extended Data Fig. 1b. **b**, Cryo-EM reconstructions of amyloid filaments from individuals 1–4 with FTLD–FET showing a readily traceable protein backbone and well-resolved amino acid side-chain densities. All four reconstructions have an identical filament fold. Resolution estimates are indicated. Scale bars, 2 nm.

filaments from individual 4, which confirmed their identity and showed that they had the type I TMEM106B fold[38] (Extended Data Fig. 2 and Extended Data Table 2). TMEM106B filaments were not observed in individual 1. In agreement with this finding, using mass spectrometry, peptides from the region forming the TMEM106B filament core were detected in insoluble extracts from individual 4 and an aged neurologically normal individual, but not from individual 1 (Extended Data Fig. 3). Individual 1 died at 30 years of age whereas the other individuals were over 50 years old at death, consistent with the previously reported age-dependent accumulation of TMEM106B filaments in human brain[38,41]. TMEM106B filaments were previously shown to accumulate in neurologically normal brains, as well as in the presence of neurodegenerative disease-characteristic amyloid filaments, with no clear relationship between disease status and TMEM106B filament fold[38,41,42].

The cryo-EM images yielded 112,000–358,000 segments of non-TMEM106B filaments per individual (Extended Data Table 2). Reference-free two-dimensional (2D) classification showed the presence of a single predominant filament population for all individuals, with a helical cross-over spacing of approximately 405 Å (Extended Data Fig. 4a). The 2D classes of this predominant filament population did not correspond to any known amyloid filament structure. The 2D classes for individual 4 also showed the presence of amyloid-β 42 (Aβ42) filaments (Extended Data Fig. 2), consistent with sparse amyloid-β plaques in the prefrontal cortex of this individual (Extended Data Table 1). In agreement, peptides corresponding to Aβ42 were identified by mass spectrometry in the insoluble extracts from individual 4, but not from either individual 1 or a neurologically normal individual (Extended Data Fig. 3). We determined the cryo-EM structure of the Aβ42 filaments from individual 4 (Extended Data Fig. 2 and Extended Data Table 2), which showed that they had the type II Aβ42 filament fold as previously found in individuals exhibiting amyloid-β plaque copathology, including in FTLD with TDP-43 and tau pathology[37].

### Structure of TAF15 filaments in FTLD–FET

For the predominant, unassigned filament population we generated de novo initial three-dimensional (3D) maps from well-resolved 2D classes using the sinogram approach detailed in ref. 43 (Extended Data Fig. 4b,c). Helical processing of each individual dataset yielded four superimposable maps at resolutions of between 2.0 and 2.7 Å (Fig. 2b and Extended Data Table 2). The protein backbone and amino acid side-chains were unambiguously resolved in our cryo-EM

reconstructions (Extended Data Fig. 5), thereby identifying the filament-forming protein by its sequence. Contrary to our initial expectation, the filaments are formed from TAF15 and not FUS. Our mass spectrometry analysis of insoluble extracts from individuals 1 and 4, and from the neurologically normal individual, corroborated this finding. Whereas peptides mapping to FUS and TAF15 were detected for all individuals, only those mapping to the region forming the core of TAF15 filaments could distinguish between the individuals with FTLD–FET and the neurologically normal individual (Extended Data Fig. 3). These results suggest that the formation of TAF15 amyloid filaments characterizes FTLD–FET, in analogy to TDP-43 and tau amyloid filaments that characterize other types of FTLD[3,5].

The TAF15 filaments comprise a single protofilament with a left-handed helical twist, with the ordered filament fold formed by residues 7–99 of TAF15, which are part of its LCD (Fig. 3a,b). Perpendicular to the helical axis, the shape of the filament fold somewhat resembles a scooter, with the proximal N and C termini representing the two handlebars (Fig. 3c). The filament fold contains 13 β-strands of between two and eight residues in length, which encompass 57% of all residues. These β-strands, together with their counterparts in adjacent TAF15 molecules, form parallel, in-register β-sheets characteristic of amyloid filaments (Extended Data Fig. 6a,b). Eight of the β-sheets are stacked with interdigitated side-chains, forming zipper packing typical of amyloid filaments[44] (Fig. 3c). Viewed along the helical axis, the termini of each TAF15 molecule are on different planes and interact with the molecules above and below (Extended Data Fig. 6b).

The fold is enriched in glycine, tyrosine, glutamine and serine residues, each contributing 16–24% of all residues. Glycine residues mainly facilitate turns between β-strands (Extended Data Fig. 7a). Among the tyrosine residues, all non-solvent-exposed side-chains are hydrogen bonded in the model (Extended Data Fig. 7b). In addition, the off-centred, parallel orientation of their aromatic rings, with a distance of 3.2–3.5 Å between aromatic planes, allows for staggered stacking interactions (Extended Data Fig. 7c). The abundant glutamine residues, together with the six asparagine residues, engage in extended hydrogen-bonding networks (Extended Data Fig. 7d,e). The majority of their side-chain amide groups form hydrogen-bonded ladders with their counterparts in neighbouring TAF15 molecules, as often observed in amyloid filaments. The side-chain amides also form intra- and intermolecular hydrogen bonds with each other, main-chain carbonyl groups and ordered solvent. Further hydrogen bonding is provided by the abundance of serine residues, in addition to three

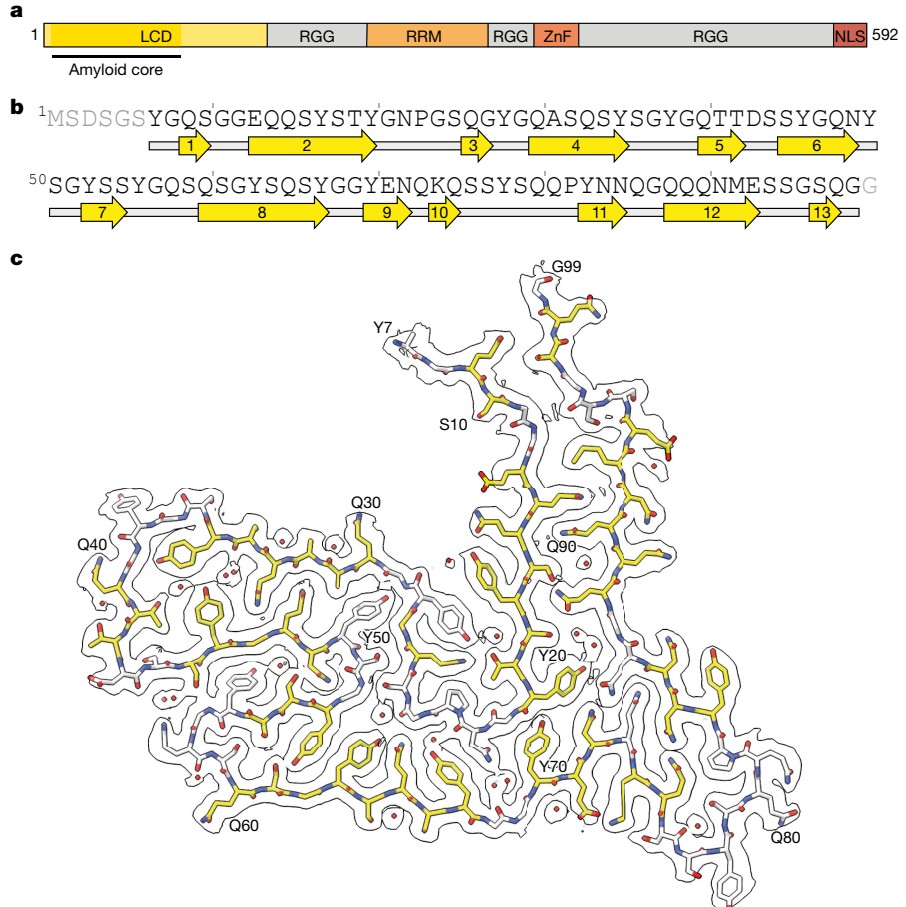

**Fig. 3 | Cryo-EM structure of TAF15 amyloid filaments from FTLD–FET.**
**a**, Domain organization of TAF15. The region comprising the ordered core of TAF15 amyloid filaments is indicated. RRM, RNA recognition motif; ZnF, zinc finger domain. **b**, Sequence alignment of secondary structure elements of the TAF15 amyloid filament fold. Arrows indicate β-strands. **c**, Cryo-EM reconstruction and atomic model of the TAF15 amyloid filament structure, shown for a single TAF15 molecule perpendicular to the helical axis. The carbon atoms of residues forming β-strands are shown in yellow and ordered solvent as red spheres.

threonine residues (Extended Data Fig. 7f). The side-chains of the four acidic residues in the fold face the solvent and one of these, E71, forms a salt bridge with the only basic residue, K74 (Extended Data Fig. 7g). Only four residues (P23, A31, P82 and M92) possess side-chains lacking polar groups. Consistent with a protein fold stabilized by intricate hydrogen-bonding networks, the high-resolution map shows that the TAF15 filament fold is well hydrated. We modelled 23 ordered water molecules per TAF15 molecule, each contributing between two and four hydrogen bonds with either polar amino acid side-chains, the backbone or other ordered solvent molecules (Fig. 3c and Extended Data Fig. 7h).

Additional densities on the filament surface that could not be modelled confidently are present in the cryo-EM maps (Extended Data Fig. 6c,d). The smaller densities may be attributed to ordered solvent whereas the larger ones might indicate the binding of other molecules. The majority of these larger densities appear connected along the filament axis, suggesting that the molecules do not follow the same helical symmetry as TAF15. The additional densities do not appear to connect to the density for TAF15 and are thus unlikely to represent covalent post-translational modifications. A planar density adjacent to a flat surface formed by residues 60–64 is reminiscent of those seen in TDP-43 filaments from human brain[4,5]. There is also an external density adjacent to the side-chain of Y83. Two densities are located in grooves along the filament axis formed by residues 85–89 and 94–98, with the orientation of the amino acid side-chains and

main-chain carbonyl groups of these residues allowing for hydrogen bonding.

## TAF15 filaments in motor regions

FTLD–FET is often associated with FET protein- and transportin 1-immunoreactive inclusions in upper and lower motor neurons[12,15,23,27]. Moreover, individuals 1 and 4 exhibited upper and lower motor neuron pathology and individual 4 had received a clinical diagnosis of probable ALS before being diagnosed with FTD (Extended Data Fig. 8a,b, Extended Data Table 1 and Methods). We therefore performed immunohistochemistry on the spinal cord, motor cortex and brainstem of the four individuals using antibodies against FET proteins and transportin 1. FUS-, TAF15- and transportin 1-immunoreactive inclusions were readily observed in upper and lower motor neurons for individuals 1 and 4, but were sparse or absent for individuals 2 and 3 (Fig. 4a and Extended Data Fig. 8c–e). We did not detect EWS-immunoreactive motor neuron inclusions.

To investigate the presence, identities and structures of amyloid filaments in motor neuron inclusions of individuals with motor neuron pathology we determined the cryo-EM structures of filaments extracted from the motor cortex of individual 1 and from the medulla of individual 4. For both individuals we found abundant TAF15 filaments with the same fold as detected previously in the prefrontal and temporal cortices (Fig. 4b and Extended Data Table 2). We did

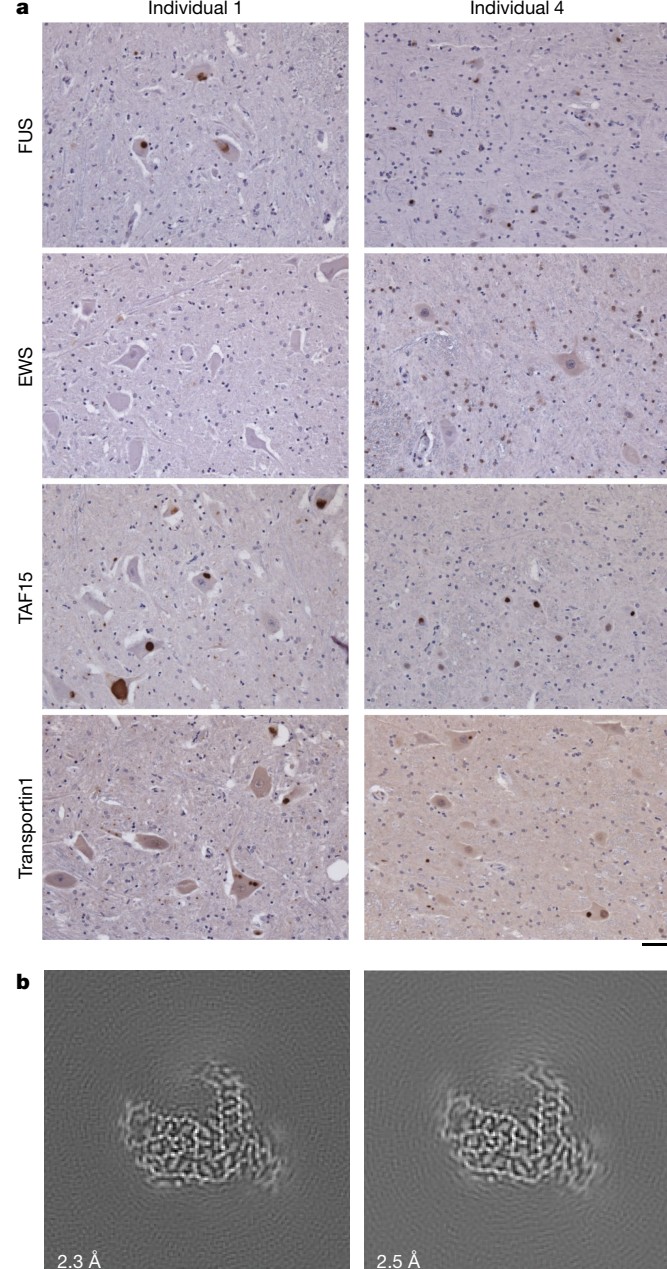

**a**

Individual 1 | Individual 4

FUS
EWS
TAF15
Transportin 1

**b**

2.3 Å | 2.5 Å

**Fig. 4 | Motor neuron inclusions and TAF15 filaments. a**, FUS, EWS, TAF15 and transportin 1 immunoreactivity (brown) in the spinal cord of individuals 1 and 4 with FTLD–FET. Sections were counterstained with haematoxylin (blue). Motor neuron inclusions were immunoreactive for FUS, TAF15 and transportin 1. Antibodies against EWS showed only diffuse labelling of nuclei. Results are representative of $n \geq 3$ technical replicates per individual. Additional immunohistochemistry of spinal cord, motor cortex and brainstem for all four individuals is shown in Extended Data Fig. 8. Scale bar, 50 μm. **b**, Cryo-EM reconstructions of amyloid filaments from the motor cortex of individual 1 (left) and medulla of individual 4 (right), showing TAF15 filaments with a fold identical to those from prefrontal cortices. Resolution estimates are indicated. Scale bar, 2 nm.

not find filaments of FUS. For individual 4 we also found TMEM106B and Aβ42 filaments with the same folds as in the prefrontal cortex, in addition to TAF15 filaments (Extended Data Fig. 9 and Extended Data Table 2). No TMEM106B or Aβ42 filaments were found for individual 1, mirroring our results from the prefrontal cortex. These results suggest that TAF15 form amyloid filaments in upper and lower motor

neuron inclusions in FTLD–FET associated with upper and lower motor neuron pathology.

## Discussion

While amyloid assembly of TDP-43 or tau is the hallmark of the majority of cases of FTLD, the assembled protein that characterizes the remaining approximately 10% of cases, termed FTLD–FET, was previously unknown. Using cryo-EM, we found abundant TAF15 amyloid filaments with a shared filament fold in the brain of four individuals with FTLD–FET. In one individual we uniquely detected TAF15 filaments in the absence of Aβ and TMEM106B filaments. TAF15-immunoreactive inclusions and TAF15 filaments have not previously been observed in other neurodegenerative conditions, or in neurologically normal individuals[23–27,38]. Together these results suggest that the formation of TAF15 amyloid filaments characterizes FTLD–FET, thereby adding TAF15 to the small group of proteins that form amyloid filaments associated with neurodegenerative disease alongside proteins such as tau, TDP-43 and α-synuclein[45].

The presence of TAF15 amyloid filaments is consistent with the immunoreactivity of neuronal cytoplasmic inclusions against TAF15 (refs. 23,26,27) (Fig. 1a), as well as with the propensity of TAF15 to form amyloid filaments in vitro[29,31,32]. We did not find amyloid filaments of FUS, despite the inclusions also exhibiting immunoreactivity against this protein[12,15] (Fig. 1a) and its propensity to also form filaments in vitro[18–20]. This was supported by the ability of mass spectrometry to discriminate between individuals with FTLD–FET and a neurologically normal individual by the presence of peptides from the TAF15 filament core, but not by FUS peptides (Extended Data Fig. 3).

Three non-mutually exclusive scenarios may account for the immunoreactivity of the inclusions against FUS in the absence of FUS filaments. First, filaments of FUS may be present in significantly smaller numbers than those of TAF15, thereby escaping our cryo-EM analysis. However, we note that cryo-EM structures of filaments accounting for only a few percent of the total population have been determined using similar approaches[3,46].

Second, FUS may form filaments that were not captured by our extraction method, possibly because of differences in stability or solubility, although we did not find filaments in the sarkosyl-soluble brain fraction (Extended Data Fig. 1a). This scenario would imply that putative FUS filaments behave differently to filaments of tau, α-synuclein, TDP-43 and Aβ, which characterize most cases of neurodegenerative disease and which were all previously shown to be extracted from human brain using the method used in this study[3,5,36–38]. We also note that all of these neurodegenerative diseases are characterized by intracellular amyloid filament inclusions of only one protein.

Third, non-filamentous FUS may be present in inclusions, as amyloid filament inclusions are known to sequester non-filamentous proteins[47]. The overlapping protein and RNA interactions of FET proteins suggest that TAF15 amyloid filaments may sequester non-filamentous FUS[28], which may also account for reports of occasional inclusion immunoreactivity against EWS[23,26,27]. Similarly, the immunoreactivity of the inclusions against transportin 1 may be due to its interaction with the NLS of TAF15 (refs. 24,25) (Fig. 1a), which lies outside of the ordered core of the filaments (Fig. 3a). Future development of experimental models that reproduce the TAF15 amyloid filament structure identified in this work will enable testing of this hypothesis.

We also found abundant TAF15 amyloid filaments with the same filament fold in the motor cortex and brainstem of two of the individuals (Fig. 4b), one of whom had received a clinical diagnosis of probable ALS before developing FTD. These individuals showed upper and lower motor neuron pathology and had FUS-, TAF15- and transportin 1-immunoreactive inclusions in upper and lower motor neurons (Fig. 4a and Extended Data Fig. 8). The other two individuals lacked motor neuron pathology and had scarce or absent motor neuron

inclusions. These results suggest that the formation of TAF15 amyloid filaments can be associated with motor neuron pathology and may underlie a disease spectrum of FTLD and motor neuron disease. This may be analogous to FTLD and ALS with TDP-43 inclusions, which share a distinct TDP-43 filament fold[5]. In support of this hypothesis, cases of sporadic ALS with FET protein- and transportin 1-immunoreactive inclusions in the absence of FTD have been reported[48,49]. Future studies should examine the identities and structures of amyloid filaments from these cases to further test this hypothesis.

Rare mutations in the genes that encode TDP-43 and tau give rise to amyloid filament assembly and inherited FTLD[6–10]. Rare mutations in the gene that encodes the filament-forming protein in FTLD–FET would, therefore, be expected give rise to inherited forms of this disorder. Mutations in *FUS* have not been linked to FTLD[21,22] whereas genetic studies of *TAF15* in FTLD have not been reported. Mutations in *TAF15* have been described in ALS, but their pathogenicity has not been confirmed[29,50,51]. One of these mutations, A31T, is within the region that forms the TAF15 filament core and might stabilize the filament fold by the introduction of additional hydrogen bonds with the adjacent Q48 (Fig. 3c). Our finding that TAF15 forms filaments in individuals with FTLD–FET, including those with upper and lower motor neuron pathology, should motivate genetic analysis of patient cohorts to elucidate the potential contribution of rare *TAF15* mutations to FTLD and motor neuron disease.

Rare mutations in *FUS* can cause ALS[16,17]. In these cases, because motor neuron inclusions are immunoreactive against FUS but not TAF15, EWS or transportin 1 (refs. 23,25), it is unlikely that TAF15 forms amyloid filaments but possible that these inclusions may contain FUS amyloid filaments. This supports the hypothesis that the disease mechanisms of ALS caused by *FUS* mutations are distinct from those of FTLD–FET and sporadic ALS with FET protein-immunoreactive inclusions, as previously suggested[23,48,49]. Additional evidence for different disease mechanisms comes from the observation that FUS is hypomethylated in FTLD–FET but not in cases of ALS caused by *FUS* mutations[52]. Future studies should focus on investigating the presence, identities and structures of amyloid filaments in cases of this rare form of familial ALS.

We found that a single TAF15 filament fold characterized individuals with FTLD–FET in this study (Fig. 2b). For TDP-43, tau and α-synuclein, distinct filament folds characterize different neurodegenerative disorders[45]. Two additional rare neurodegenerative disorders—neuronal intermediate filament inclusion body disease and basophilic inclusion body disease—also present with inclusions that are immunoreactive against FET proteins and transportin 1 (refs. 23,26,27). Potentially, distinct filament folds of TAF15 may underlie these disorders.

## Conclusion

The presence of abundant TAF15 amyloid filaments with a characteristic fold in FTLD establishes TAF15 as a member of the small group of proteins known to form neurodegenerative disease-associated amyloid filaments. This focuses attention on the role of TAF15 proteinopathy in neurodegenerative disease. In accordance with consensus recommendations for FTLD nomenclature[53], we strongly advocate that the frequently used term FTLD–FUS should be abandoned in favour of the previously suggested FTLD–FET[23], and that it may even be appropriate to consider the use of the term FTLD–TAF. The structures of TAF15 filaments will guide the development of model systems to enable studies of disease mechanisms, and will provide a basis for the design of diagnostic and therapeutic tools targeting TAF15 proteinopathy in neurodegenerative disease.

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

## Methods

### Neuropathological analyses

Human tissue samples were from the Dementia Laboratory Brain Library at Indiana University School of Medicine (individuals 1 and 4) and from the Queen Square Brain Bank for Neurological Disorders at University College London Queen Square Institute of Neurology (individuals 2 and 3). Their use in this study was approved by the ethical review processes at each institution. Informed consent was obtained from patients' next of kin.

These individuals were selected based on a neuropathological diagnosis of FTLD–FET. All individuals had abundant neuronal cytoplasmic inclusions, as well as sparse neuronal intranuclear inclusions and dystrophic neurites, in the prefrontal and temporal cortices. The inclusions were immunoreactive against ubiquitin, p62, FUS, TAF15 and transportin 1. Immunohistochemistry for α-internexin did not show any neuronal intermediate filament inclusions. Haematoxylin and eosin staining did not show any basophilic inclusions. These results are consistent with the FTLD subtype atypical FTLD with ubiquitin-positive inclusions (aFTLD-U)[2].

Individuals 1 and 4 also exhibited upper and lower motor neuron pathology, both showing loss of upper and lower motor neurons. Luxol fast blue staining of myelin also showed extensive lateral and anterior corticospinal tract loss in the thoracic and lumbar spinal cord for individual 1. Corticospinal tract loss could not be assessed for individual 4 owing to a lack of available spinal cord. Individuals 1 and 4 also had abundant FUS-, TAF15- and transportin 1-positive cytoplasmic inclusions in upper and lower motor neurons in the motor cortex, brainstem and spinal cord.

All individuals received a clinical diagnosis of behavioural variant FTD. Individual 4 also received a diagnosis of probable ALS based on electromyography results, lower extremity hyper-reflexia and bulbar symptoms. Additional clinicopathological details are given in Extended Data Table 1.

### Genetic analyses

Whole-exome sequencing target enrichment was performed using the SureSelectTX human all-exon library (v.6, 58 megabase pairs, Agilent) and high-throughput sequencing was carried out using a HiSeq 4000 (sx75 base-pair, paired-end configuration, Illumina). Data corresponding to the coding regions of genes previously reported as being associated with FTLD were screened for potential pathogenic variants; these genes included *CHMP2B*, *EWS*, *FUS*, *GRN*, *INA*, *MAPT*, *MATR3*, *OPTN*, *SQSTM1*, *TAF15*, *TARDBP*, *TBK1*, *TIA1*, *TMEM106B*, *UBQLN1* and *VCP*. Analysis of *C9orf72* for hexanucleotide repeat expansion was performed using repeat-primed PCR as previously described[54]. No mutations associated with FTLD were found, and the individuals had wild-type *FUS*, *TAF15* and *EWS*

### Extraction of sarkosyl-insoluble proteins

Sarkosyl-insoluble proteins were extracted from flash-frozen tissue (prefrontal, temporal and motor cortices, and medulla) as previously described[5]. Grey matter was dissected and homogenized using a Polytron (Kinematica) in 40 volumes (v/w) of extraction buffer (10 mM Tris-HCl pH 7.4, 0.8 M NaCl, 10% sucrose, 1 mM dithiothreitol and 1 mM EGTA) containing protease and phosphatase inhibitor cocktail (Pierce). A 25% solution of sarkosyl in water was added to homogenates to achieve a final concentration of 2% sarkosyl. Homogenates were then incubated for 1 h or overnight at 37 °C with orbital shaking at 200 rpm, followed by centrifugation at 27,000*g* for 10 min. Supernatants were retained and centrifuged in 1 ml aliquots at 166,000*g* for 20 min. Each pellet was soaked in 20 µl of extraction buffer containing 1% sarkosyl at 37 °C for at least 20 min and then resuspended by pipetting. Six pellets were combined, topped up to 0.5 ml with the same buffer and sonicated for 5 min at 50% amplitude (Qsonica Q700). Samples were

then diluted to 1 ml with the same buffer and centrifuged at 17,000*g* for 5 min. Supernatants were retained and centrifuged at 166,000*g* for 20 min. Pellets were soaked and resuspended as before. Samples were topped up to 1 ml with extraction buffer containing 1% sarkosyl and incubated overnight at 37 °C with orbital shaking at 200 rpm. Samples were centrifuged at 166,000*g* for 20 min and pellets resuspended in 25 µl g$^{-1}$ tissue of 20 mM Tris-HCl pH 7.4 and 150 mM NaCl by soaking at 37 °C, pipetting and sonication for 5 min at 50% amplitude. 1–2 g of tissue was used for each cryo-EM sample. All centrifugation steps were carried out at 25 °C.

### Immunolabelling

For histology, brain hemispheres were fixed with 10% buffered formalin and embedded in paraffin. Deparaffinized sections (8 µm thickness) were treated with 88% formic acid for 5 min and incubated in 10 mM sodium citrate buffer at 105 °C for 10 min. After washing, sections were treated with non-fat dry milk in Tris-buffered saline and then incubated overnight with primary antibodies against either FUS (Proteintech, no. 11570-1-AP at a dilution of 1:1,000), TAF15 (Bethyl, no. IHC-00094 at a dilution of 1:500), EWS (Santa Cruz, no. sc-28327 at a dilution of 1:250) or transportin 1 (abcam, no. ab10303 at a dilution of 1:200) in Tris-buffered saline. Following incubation with biotinylated secondary antibodies overnight, labelling was detected using the ABC staining kit (Vector) with 3,3'-diaminobenzidine. Sections were counterstained with haematoxylin.

For immunoblotting, samples were resolved using 4–12% BIS-Tris gels (Novex) at 200 V for 40 min and transferred to nitrocellulose membranes. Membranes were blocked in PBS containing 1% bovine serum albumin and 0.2% Tween for 30 min at room temperature and incubated with primary antibodies against either FUS (Proteintech, no. 11570-1-AP at a dilution of 1:5,000), TAF15 (Bethyl, no. A300-308A at a dilution of 1:5,000), EWS (Santa Cruz, no. sc-28327 at a dilution of 1:250) or transportin 1 (abcam, no. ab10303 at a dilution of 1:500) for 1 h at room temperature. Membranes were then washed three times with PBS containing 0.2% Tween and incubated with either Goat Anti-Mouse IgG StarBright Blue 700 (Bio-Rad) or Anti-Rabbit IgG DyLight 800 4× PEG Conjugate (Cell Signaling Technology) secondary antibodies for 1 h at room temperature. Membranes were then washed three times with PBS containing 0.2% Tween and imaged using a ChemiDoc MP (Bio-Rad).

### Mass spectrometry

Sarkosyl-insoluble proteins were extracted from 0.2 g of grey matter. The final pellet was dried by vacuum centrifugation (Savant) then soaked in 20 µl of hexafluoroisopropanol, incubated at 37 °C for 1 h, resuspended and topped up to 100 µl of solvent. Samples were sonicated three times for 3 min each at 50% amplitude in a water bath (QSonica Q700). Any non-disassembled filaments were removed by centrifugation at 166,000*g* for 30 min. The supernatant was dried by vacuum centrifugation.

Dried protein samples were resuspended in 8 M urea and 50 mM ammonium bicarbonate, reduced with 5 mM dithiothreitol and alkylated with 10 mM iodoacetamide. Samples were diluted to 1 M urea with 50 mM ammonium bicarbonate and incubated with chymotrypsin (Promega) overnight at 25 °C. Digestion was stopped by the addition of formic acid to a final concentration of 0.5%, followed by centrifugion at 16,000*g* for 5 min. Supernatants were desalted using home-made C18 stage tips (3M Empore) packed with Oligo R3 resin (Thermo Fisher Scientific) resin. Bound peptides were eluted with 5–60% acetonitrile in 0.5% formic acid and partially dried in a Speed Vac (Savant).

Peptide mixtures were analysed by liquid chromatography–tandem mass spectrometry using an Ultimate 3000 RSLCnano system (Thermo Fisher Scientific) coupled to an Orbitrap Q Exactive HFX mass spectrometer (Thermo Fisher Scientific). Peptides were trapped using a 100 µm × 2 cm, PepMap100 C18 nanotrap column (Thermo Fisher Scientific) and separated on a 75 µm × 50 cm, EASY-Spray HPLC Column using

a binary gradient consisting of buffer A (2% acetonitrile, 0.1% formic acid) and buffer B (80% acetonitrile, 0.1% formic acid) at a flow rate of 300 nl min⁻¹ for 190 min. For data independent acquisition, MS1 spectra were acquired at a resolution of 60,000, mass range 385–1,015 $m/z$ and maximum injection time 60 ms. MS2 analysis was carried out at a resolution of 15,000, and 25 MS2 scans with 24 $m/z$ isolation window.

Liquid chromatography–tandem mass spectrometry data were processed using DIA-NN software (v.1.8.1) in library-free mode[55]. The sequence database was automatically generated from the UP000005640_9606 human proteome fasta file (March 2023). Two chymotrypsin missed cleavages were allowed in the search parameters. Carbamidomethyl cysteine was set as static modification and methionine oxidation as variable modification. Precursor mass range was set as 370–1,100 $m/z$ and default settings were used for other parameters. The files report.pg_matrix.tsv (protein) and report.pr_matrix.tsv (peptide) were used for analyses.

### Negative-stain electron microscopy
Sarkosyl-insoluble proteins were extracted from 0.1 g of grey matter, the supernatant of the first ultracentrifugation being retained as the sarkosyl-soluble fraction. The final pellet of sarkosyl-insoluble protein was resuspended in 50 μl of 20 mM Tris-HCl pH 7.4 and 150 mM NaCl. Both sarkosyl-soluble and -insoluble fractions were sonicated for 5 min at 50% amplitude. Samples were diluted up to tenfold in 20 mM Tris-HCl pH 7.4 and 150 mM NaCl. Glow-discharged 400-mesh carbon-coated copper grids (Electron Microscopy Sciences) were incubated face-down on 4 μl of sample for 1 min. Grids were washed three times using Millipore-filtered water and once using 2% uranyl acetate, then stained for 30 s in 2% uranyl acetate before blotting with filter paper. Dried grids were imaged with a 120 keV Tecnai Spirit microscope (Thermo Fisher Scientific) with an Orius CCD detector (Gatan).

### Cryo-EM
Extracted sarkosyl-insoluble proteins were incubated with 0.4 mg ml⁻¹ pronase (Sigma) for 1 h at room temperature and centrifuged at 3,000$g$ for 15 s to remove large debris. Supernatants were retained and applied to glow-discharged 1.2/1.3 μm holey carbon-coated 200-mesh gold grids (Quantifoil) and plunge-frozen in liquid ethane using a Vitrobot Mark IV (Thermo Fisher Scientific). Images were acquired using a 300 keV Titan Krios microscope (Thermo Fisher Scientific) with either a Falcon 4 detector (Thermo Fisher Scientific) or a K3 detector (Gatan) and GIF-quantum energy filter (Gatan) operated at a slit width of 20 eV. Aberration-free image shift within the EPU software (Thermo Fisher Scientific) was used during image acquisition. Further details are given in Extended Data Table 2.

### Helical reconstruction
Movie frames were gain-corrected, aligned, dose-weighted and summed using the motion correction programme in RELION-4.0 (ref. 56). Motion-corrected micrographs were used to estimate contrast transfer function (CTF) using CTFFIND-4.1 (ref. 57). All subsequent image processing was performed using helical reconstruction methods in RELION-4.0 (refs. 58,59). Amyloid filaments were picked manually, and reference-free 2D classification was performed to remove suboptimal segments. Initial 3D reference models were generated de novo by producing sinograms from 2D class averages as previously described[43]. Masked 3D autorefinements with optimization of helical twist were performed, followed by iterative Bayesian polishing and CTF refinement[56,60]. Where beneficial, 3D classification was used to further remove suboptimal segments; 3D autorefinement, Bayesian polishing and CTF refinement were then repeated. Final reconstructions were sharpened using the standard post-processing procedures in RELION-4.0, and overall resolutions were estimated from Fourier shell correlations of 0.143 between the two independently refined half-maps using phase-randomization to correct for convolution effects

of a generous, soft-edged solvent mask[61]. Local-resolution estimates were obtained using the same phase-randomization procedure but with a soft spherical mask that was moved over the entire map. Helical symmetry was imposed using the RELION Helix Toolbox. Further details are given in Extended Data Table 2.

### Atomic model building and refinement
The atomic models were built de novo and refined in real space in COOT[62] using the best-resolved map. Rebuilding using molecular dynamics was carried out in ISOLDE[63]. The model was refined in Fourier space using REFMAC5 (ref. 64), with appropriate symmetry constraints defined using Servalcat[65]. To confirm the absence of overfitting the model was shaken, refined in Fourier space against the first half-map using REFMAC5 and compared with the second half-map. Geometry was validated using MolProbity[66]. Molecular graphics and analyses were performed in ChimeraX[67]. Model statistics are given in Extended Data Table 2.

### Reporting summary
Further information on research design is available in the Nature Portfolio Reporting Summary linked to this article.

## Data availability
Whole-exome data have been deposited in the National Institute on Ageing Alzheimer's Disease Data Storage Site (NIAGADS) under accession code NG00107. Mass spectrometry data have been deposited to the Proteomics Identifications (PRIDE) database under accession code PXD044821. Cryo-EM datasets have been deposited to the Electron Microscopy Public Image Archive (EMPIAR) under accession code nos. EMPIAR-11735 (individual 1, prefrontal cortex), EMPIAR-11736 (individual 1, motor cortex), EMPIAR-11737 (individual 2, prefrontal and temporal cortex), EMPIAR-11738 (individual 3, prefrontal and temporal cortex), EMPIAR-11739 (individual 4, prefrontal cortex) and EMPIAR-11740 (individual 4, brainstem). Cryo-EM maps have been deposited to the Electron Microscopy Data Bank under accession codes EMD-16999 and EMD-18236 (TAF15 filaments from prefrontal cortex and motor cortex, respectively, of individual 1); EMD-17022 (TAF15 filaments from prefrontal and temporal cortex of individual 2); EMD-17021 (TAF15 filaments from prefrontal and temporal cortex of individual 3); EMD-17020 and EMD-18227 (TAF15 filaments from prefrontal cortex and brainstem, respectively, of individual 4); EMD-17109 and EMD-18226 (Aβ42 filaments from prefrontal cortex and brainstem, respectively, of individual 4); EMD-18240 and EMD-18243 (singlet TMEM106B filaments from prefrontal cortex and brainstem, respectively, of individual 4); and EMD-18242 and EMD-18241 (doublet TMEM106B filaments from prefrontal cortex and brainstem, respectively, of individual 4). The atomic model for TAF15 amyloid filaments has been deposited to the Protein Data Bank (PDB) under accession code 8ONS. Atomic models of singlet and doublet TMEM106B type I filaments are available at the PDB under accession codes 7QVC and 7QVF, respectively. The atomic model of Aβ42 type II filaments is available at the PDB under accession code 7Q4M.

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

**Acknowledgements** We thank the individuals and their families for donating brain tissue; the Queen Square Brain Bank for Neurological Disorders at University College London Queen Square Institute of Neurology, which receives support from the Reta Lila Weston Institute for Neurological Studies, for supplying tissue from individuals 2 and 3; J. Grafman for supplying tissue from individual 4; M. Jacobsen for help with neuropathological examinations; R. Richardson, K. Cox and N. Maynard for help with histology and immunohistochemistry; J. Grimmett, T. Darling and I. Clayson for help with high-performance computing; K. Yamashita and G. Murshudov for help with model refinements; and T. Behr, A. Bertolotti, R. Chen, S. Davies, M. Goedert, D. Hilvert and S. Scheres for discussions. This work was supported by the electron microscopy and scientific computing facilities at the MRC Laboratory of Molecular Biology and by the Center for Medical Genomics at the Indiana University School of Medicine. This work was supported by the Medical Research Council as part of United Kingdom Research and Innovation (also known as UK Research and Innovation) (no. MC_UP_1201/25 to B.R.-F.); the US National Institutes of Health (nos. U01-NS110437, RF1-AG071177 and R01-AG080001 to R.V. and B.G.); the Alzheimer's Society (nos. AS-PG-18-004 and AS-PG-21-004 to T.L.); the Association for Frontotemporal Degeneration (no. 2019-0009 to Y.B. and T.L.); a Swiss National Science Foundation Postdoctoral Fellowship (no. P500PB_206890 to S.T.); and a Leverhulme Early Career Fellowship (no. ECF-2022-610 to D.A.). For the purposes of open access, the MRC Laboratory of Molecular Biology has applied a CC BY public copyright licence to any Author Accepted Manuscript version arising.

**Author contributions** K.L.N., L.G.A., T.L. and B.G. identified individuals. K.L.N., T.L. and B.G. performed neuropathology. Y.B., T.L. and B.G. performed immunohistochemistry. H.J.G. and R.V. performed genetic analyses. S.T. and D.A. prepared brain extracts. S.T. performed negative-stain electron microscopy. S.T. and S.Y.P.-C. performed mass spectrometry. S.T. performed immunoblot analysis. S.T. and D.A. collected cryo-EM data. S.T., A.G.M. and B.R.-F. analysed cryo-EM data. B.R.-F. supervised the study. All authors contributed to writing the manuscript.

**Competing interests** The authors declare no competing interests.

**Additional information**
**Correspondence and requests for materials** should be addressed to Benjamin Ryskeldi-Falcon.

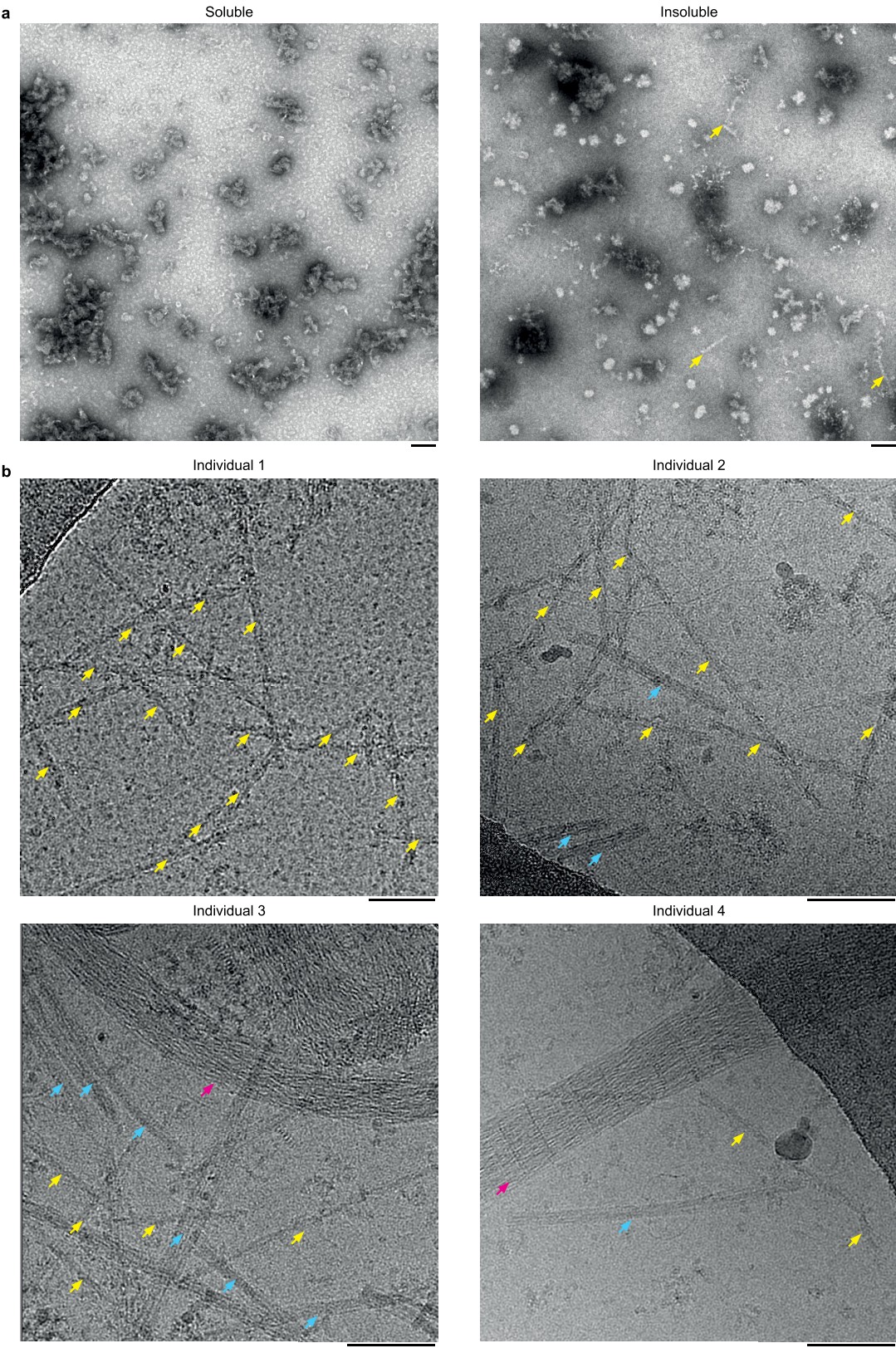

**a**
Soluble Insoluble

**b**
Individual 1 Individual 2

Individual 3 Individual 4

**Extended Data Fig. 1 | Electron micrographs of sarkosyl-soluble and -insoluble brain extracts from individuals with FTLD-FET. a**, Representative negative stain electron micrographs of the sarkosyl-soluble and -insoluble fractions of prefrontal cortex grey matter from FTLD-FET individual 1. Amyloid filaments were only observed in the insoluble fraction (yellow arrows). Scale bar, 100 nm. **b**, Cryo-EM micrographs of the sarkosyl-insoluble fraction of frontotemporal cortex grey matter from FTLD-FET individuals 1–4. Yellow arrows indicate the predominant filament population. Cyan arrows indicate TMEM106B filaments. Magenta arrows indicate collagen fibres. Scale bars, 50 nm. For **a** and **b**, the results are representative of n ≥ 3 technical replicates per individual.

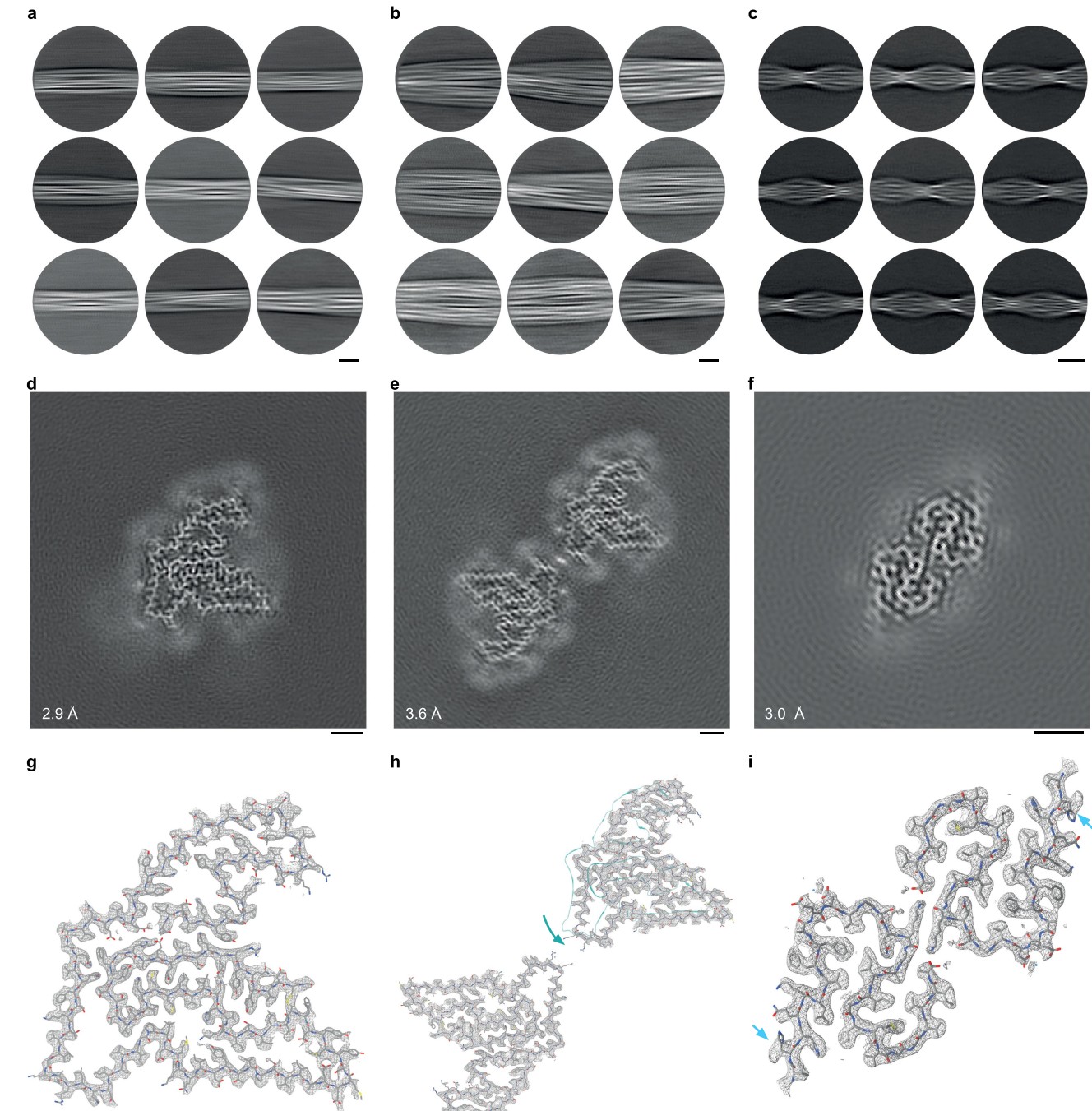

**Extended Data Fig. 2 | Cryo-EM structure of TMEM106B and Aβ42 filaments from the prefrontal cortex of FTLD-FET individual 4. a-c**, Cryo-EM reference-free 2D class averages of the filament segments used to reconstruct TMEM106B singlet (**a**), TMEM106B doublet (**b**), and Aβ42 (**c**) filaments from FTLD-FET individual 4. Scale bars, 10 nm. **d-f**, Cryo-EM reconstructions of TMEM106B singlet (**d**), TMEM106B doublet (**e**), and Aβ42 (**f**) filaments, viewed as central slices perpendicular to the helical axis. Scale bar, 2 nm. Resolution estimates are indicated. **g**, Fit of the published atomic model of singlet TMEM106B type 1 filaments (PDB-ID: 7QVC) into the density map (grey mesh), shown for a single peptide perpendicular to the helical axis. **h**, Fit of the published atomic model

of doublet TMEM106B type 1 filaments (PDB-ID: 7QVF) into the density map (grey mesh), shown for two C2 symmetry-related peptides perpendicular to the helical axis. The two chains were fit individually, as their relative orientations were rotated compared to the published model (teal). **i**, Fit of the published atomic model of Aβ42 type II filaments (PDB-ID: 7Q4M) into the reconstruction (grey mesh), shown for two C2 symmetry-related peptides perpendicular to the helical axis. The overall structure remains similar, with only subtle shifts in the backbone towards the N-termini and His13 side chain flips (cyan arrows) observed in our reconstruction.

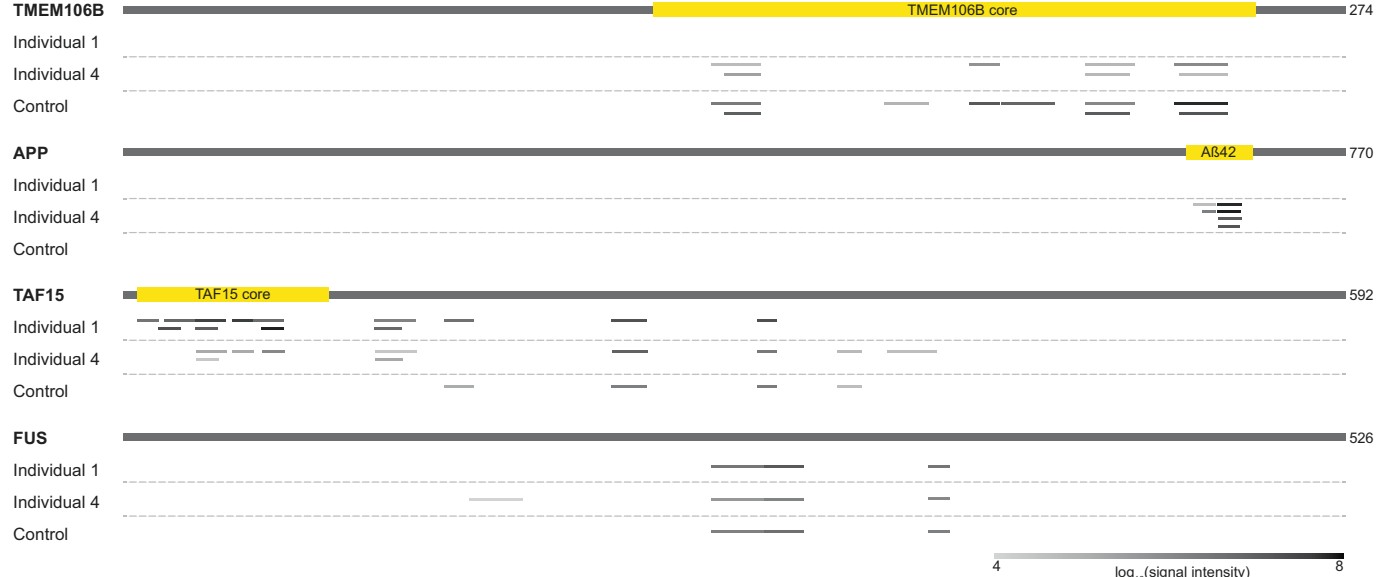

**Extended Data Fig. 3 | Mass spectrometry analysis of TMEM106B, APP, TAF15 and FUS in insoluble extracts from individuals with FTLD-FET.** Peptides identified by mass spectrometry are mapped along the protein sequences of TMEM106B, the amyloid-β precursor protein (APP), TAF15 and FUS. The amyloid filament core regions of TMEM106B and TAF15, as well as the Aβ42 peptide, are highlighted in yellow. Peptides from TMEM106B were not detected for individual 1, in agreement with the cryo-EM data. Peptides from Aβ42 were only detected for individual 4, in agreement with cryo-EM data and histopathology (Extended Data Table 1). Peptides mapping to the TAF15 filament core were only identified for individuals with FTLD-FET. No differences could be identified in FUS peptides for individuals with or without disease. Peptides from EWS and transportin-1 were not detected.

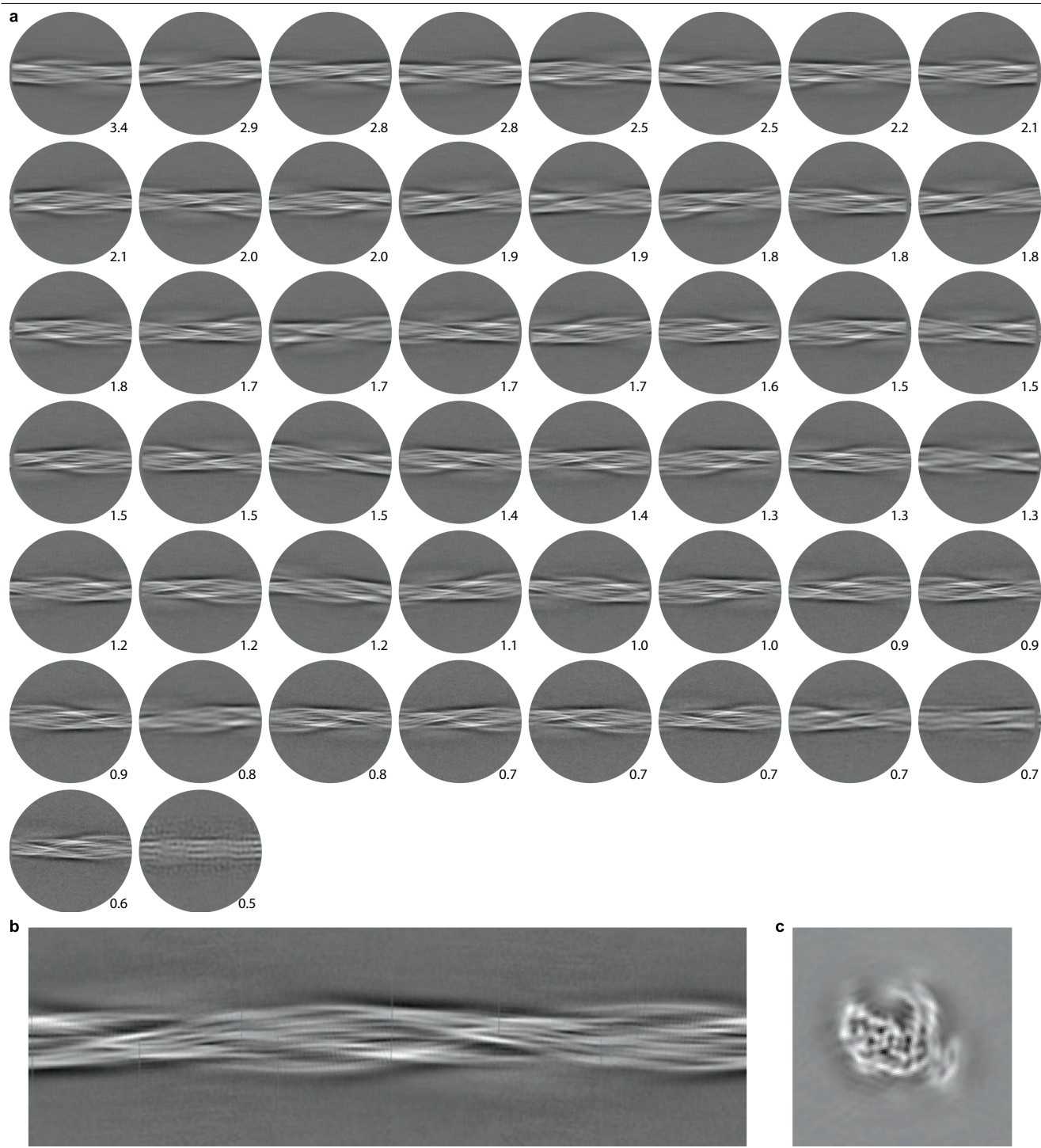

**Extended Data Fig. 4 | Cryo-EM 2D class averages and 3D initial model of amyloid filaments from FTLD-FET. a**, The 50 most populated cryo-EM reference-free 2D class averages of amyloid filaments from FTLD-FET individual 1 are shown. Numbers indicate the percentage of filament segments in each class average with respect to the total number of classified segments. The displayed classes comprise ~80% of total filament segments. The circular mask has a diameter of 400 Å and corresponds to approximately one helical cross-over of the filaments. **b,c**, An initial 3D model of amyloid filaments from FTLD-FET individual 1, generated de novo from reference-free 2D class averages, viewed as a 2D projection along the helical axis (**b**) and as a central slice perpendicular to the helical axis (**c**). Scale bars, 2 nm.

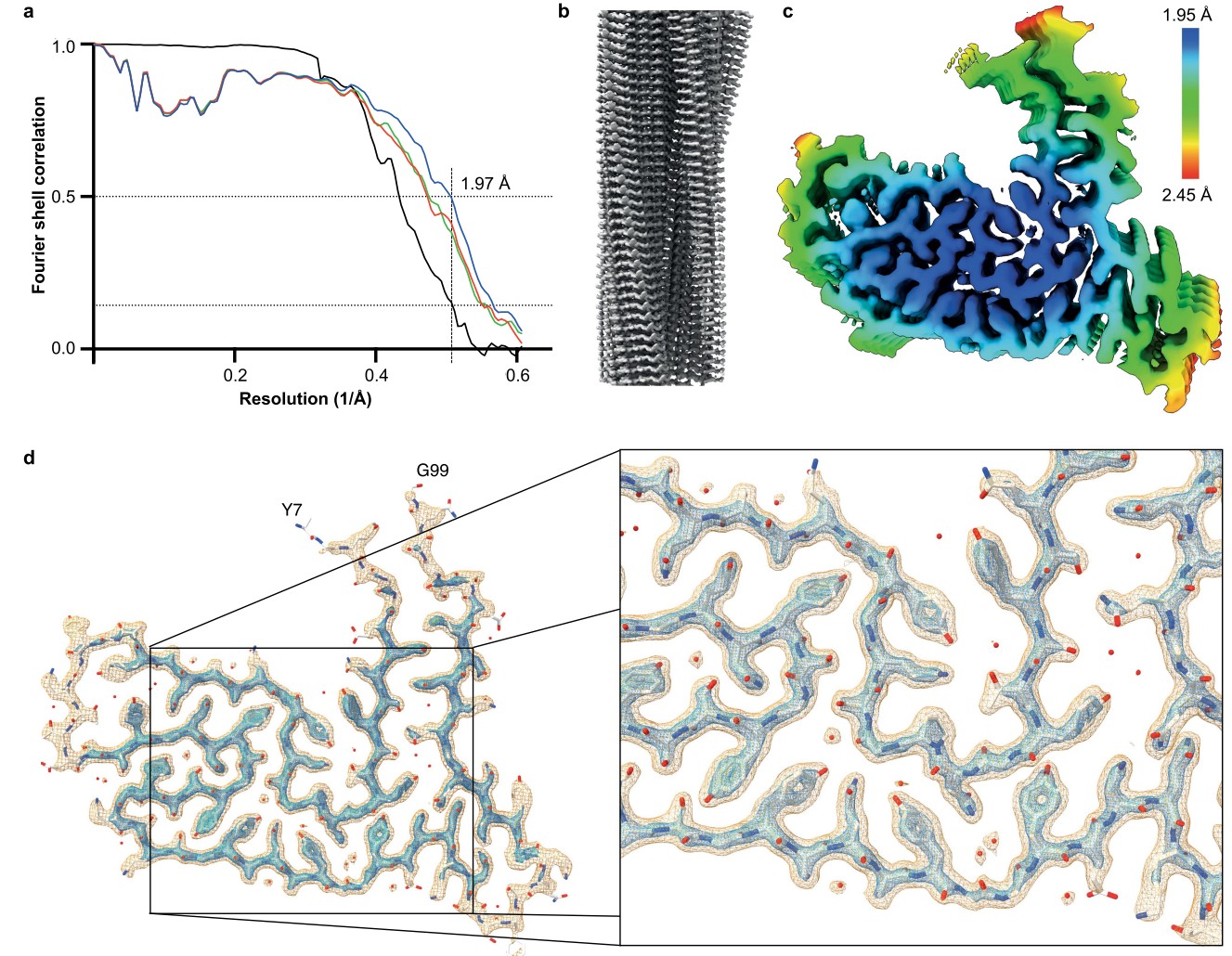

**Extended Data Fig. 5 | A high-resolution map of TAF15 amyloid filaments from FTLD-FET. a**, Fourier shell correlation (FSC) curves for two independently refined half maps from individual 1 (black line); for the refined atomic model against the cryo-EM density map (blue); and for the atomic model shaken and refined against the first (green) or second (red) independent half map. FSC thresholds of 0.143 and 0.5, as well as a vertical line at the estimated map resolution of 1.97 Å are plotted. **b**, The map viewed along the helical axis, showing well-resolved individual TAF15 molecules. **c**, The map with rainbow-coloured local resolution estimates viewed perpendicular to the helical axis. **d**, The map, shown at contour levels of 0.015 (orange) and 0.0345 (blue), viewed for a single TAF15 molecule perpendicular to the helical axis, shows a well-resolved backbone and clear side-chain densities, including aromatic rings of tyrosine residues.

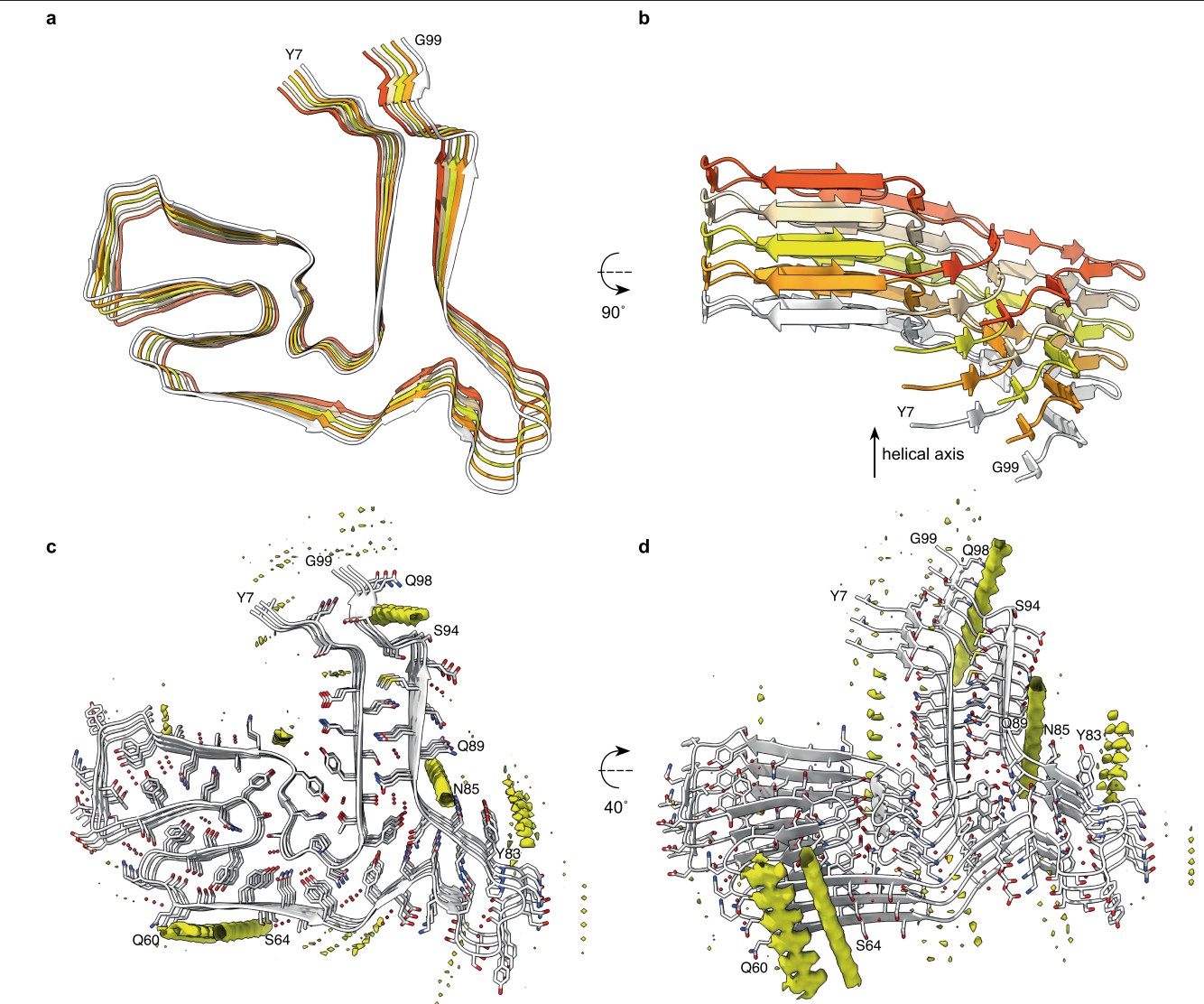

**Extended Data Fig. 6 | The TAF15 amyloid filament fold of FTLD-FET.**
**a,b**, Atomic model of TAF15 amyloid filaments from FTLD-FET shown for five differently-coloured TAF15 molecules perpendicular to (**b**) and along (**c**) the helical axis, showing that individual TAF15 molecules are not planar and that the N- and C-termini of neighbouring molecules interact with each other. **c,d**, Unmodelled densities (yellow), calculated by subtracting modelled density from the cryo-EM map, shown perpendicular to the helical axis (**d**) and rotated by 40° (**e**).

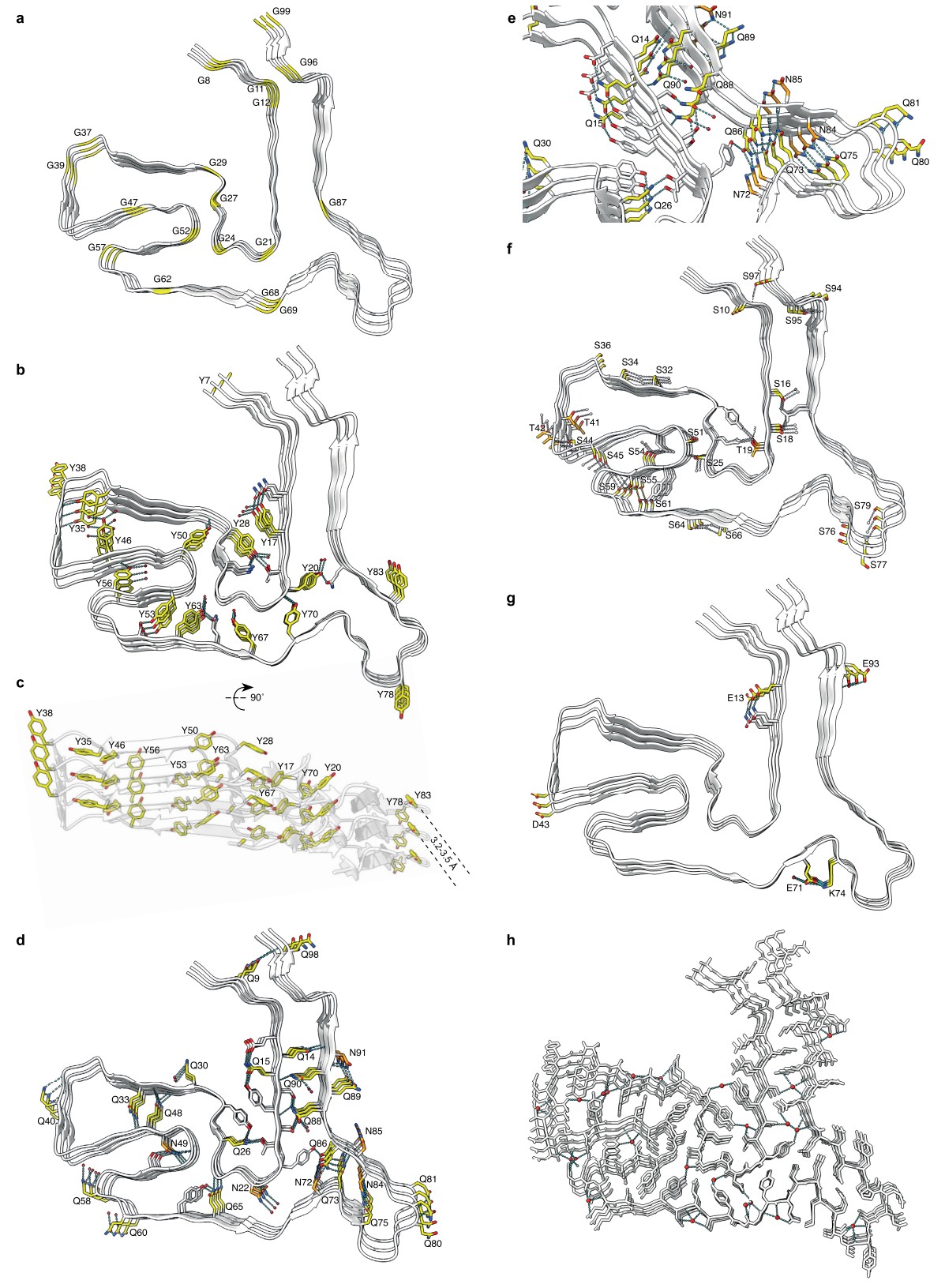

**Extended Data Fig. 7 | Structural features of the TAF15 amyloid filament fold of FTLD-FET. a–h**, Views of the atomic model of TAF15 amyloid filaments from FTLD-FET, shown for three TAF15 molecules, highlighting glycine residues (yellow) (**a**); tyrosine residues (yellow), their hydrogen bonding network (dashed lines) and staggered stacking interactions of their aromatic side chain groups (**b,c**); glutamine (yellow) and asparagine (orange) residues and their hydrogen bonding network (dashed lines) (**d,e**); serine (yellow) and threonine (orange) residues and their hydrogen bonding network (dashed lines) (**f**); charged residues (yellow), with a salt bridge between K74 and E71 (**g**); and ordered solvent molecules (red) and their hydrogen bonding network (dashed lines) (**h**).

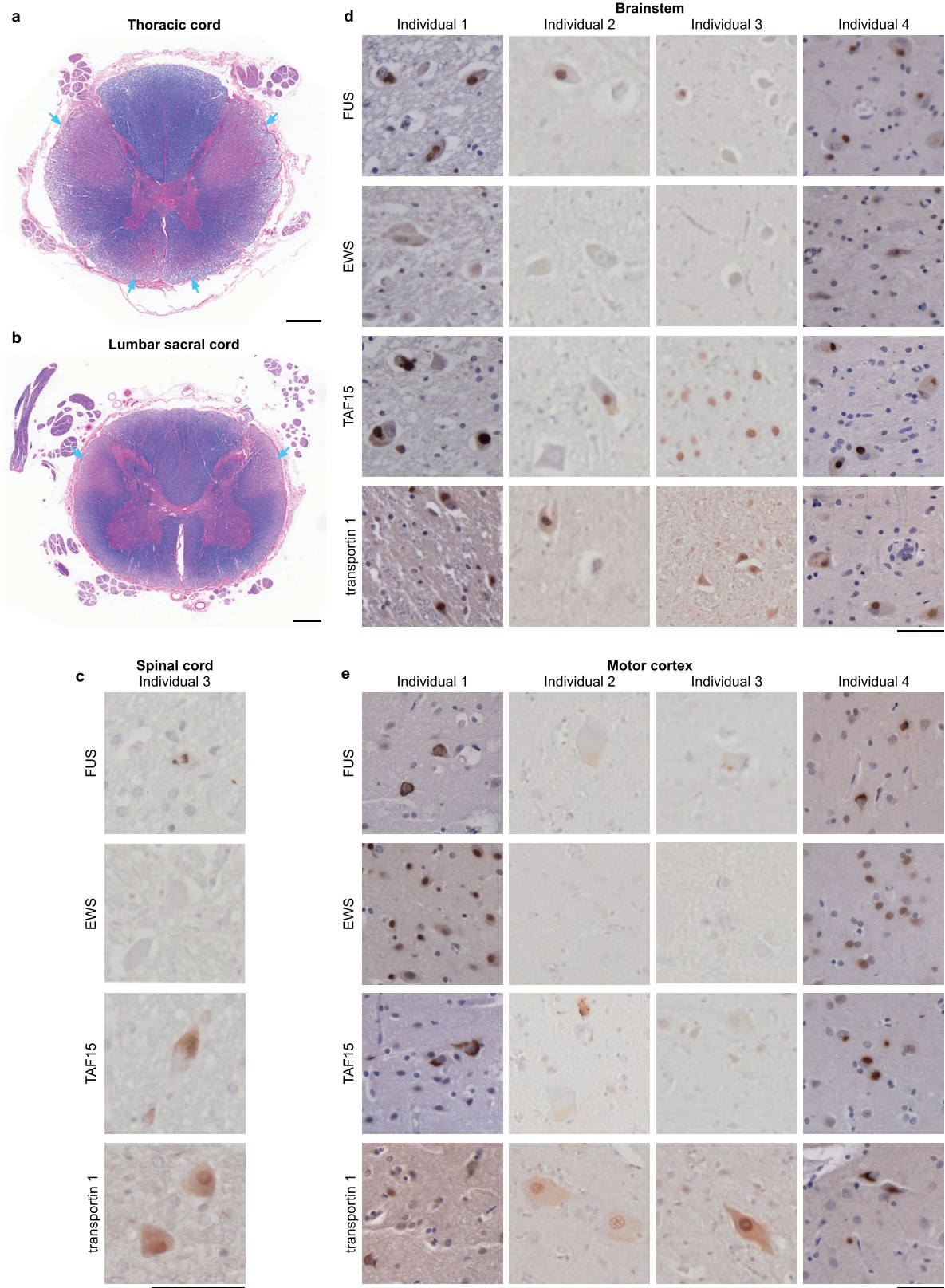

**Extended Data Fig. 8 | Motor neuron pathology. a, b,** Luxol fast blue staining of myelin in the thoracic (**a**) and lumbar (**b**) spinal cord of FTLD-FET individual 1 showing extensive corticospinal tract loss (cyan arrows). Scale bar, 1 mm. **c–e**, FUS, EWS, TAF15 and transportin 1 immunoreactivity (brown) in the spinal cord of individual 3 (**c**), the brainstem of individuals 1–4 (**d**); and the motor cortex of individuals 1–4 (**e**). Sections were counterstained with hematoxylin (blue).

Scale bars, 50 μm. Abundant motor neuron inclusions were observed for individuals 1 and 4, whereas inclusions were scarce or absent in individuals 2 and 3. Motor neuron inclusions were immunoreactive for FUS, TAF15 and transportin 1. Antibodies against EWS only showed diffuse labelling of nuclei. For **a–e**, the results are representative of n ≥ 3 technical replicates per individual.

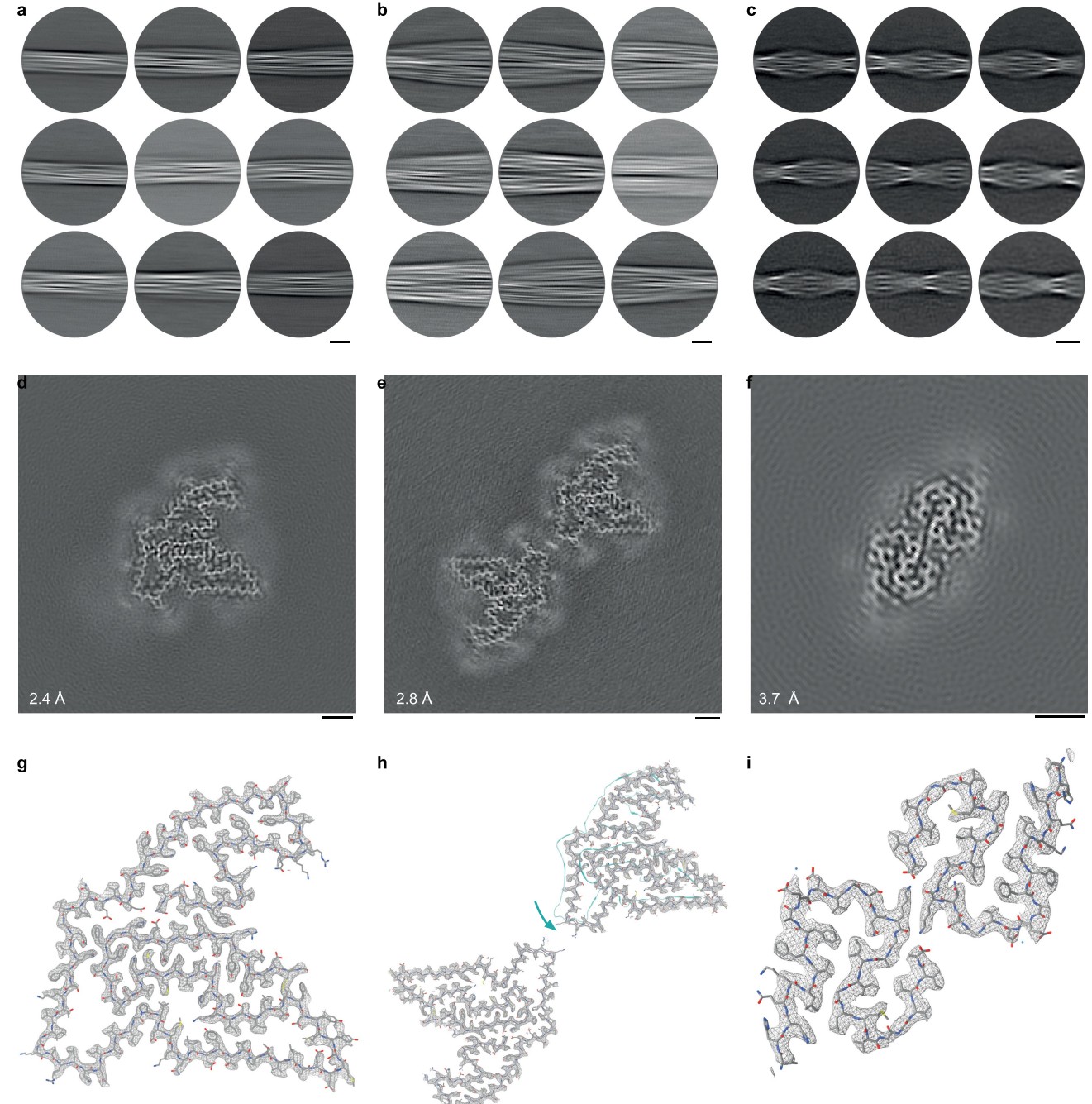

**Extended Data Fig. 9 | Cryo-EM structure of TMEM106B and Aβ42 filaments from the brainstem of FTLD-FET individual 4. a-c**, Cryo-EM reference-free 2D class averages of the filament segments used to reconstruct TMEM106B singlet (**a**), TMEM106B doublet (**b**), and Aβ42 (**c**) filaments from the brainstem of FTLD-FET individual 4. Scale bars, 10 nm. **d-f**, Cryo-EM reconstructions of TMEM106B singlet (**d**), TMEM106B doublet (**e**), and Aβ42 (**f**) filaments, viewed as a central slice perpendicular to the helical axis. Scale bar, 2 nm. Resolution estimates are indicated. **g**, Fit of the published atomic model of singlet TMEM106B type 1 filaments (PDB-ID: 7QVC) into the density map (grey mesh), shown for a single peptide perpendicular to the helical axis. **h**, Fit of the published atomic model of doublet TMEM106B type 1 filaments (PDB-ID: 7QVF) into the density map (grey mesh), shown for two C2 symmetry-related peptides perpendicular to the helical axis. The two chains were fit individually, as their relative orientations were rotated compared to the published model (teal). **i**, Fit of the published atomic model of Aβ42 type II filaments (PDB-ID: 7Q4M) into the reconstruction (grey mesh), shown for two C2 symmetry-related peptides perpendicular to the helical axis.

**Extended Data Table 1 | Clinicopathological details**

|  | Individual 1 | Individual 2 | Individual 3 | Individual 4 | Neurologically normal individual |
|---|---|---|---|---|---|
| M/F | F | M | M | M | M |
| Age (y) | 30 | 67 | 59 | 51 | 79 |
| Disease duration (y) | 9 | 6 | 10 | 6 | NA |
| Clinical diagnosis | bvFTD | bvFTD | bvFTD | probable ALS, bvFTD | neurologically normal |
| Neuropathological diagnoses | FTLD-FUS, FTLD-FET, aFTLD-U, MNP | FTLD-FUS, FTLD-FET, aFTLD-U | FTLD-FUS, FTLD-FET, aFTLD-U | FTLD-FUS, FTLD-FET, aFTLD-U, MNP | none |
| Brain regions studied | PFC, SC, MC, BS | PFC, TC, MC, BS | PFC, TC, SC, MC, BS | PFC, SC, MC, BS | PFC |
| AD-NC | A0, B0, C0 | A0, B0, C0 | A1, B0, C0 | A2, B0, C0 | A0, B1, C0 |
| α-synuclein inclusions | Absent | Absent | Absent | Absent | Absent |
| TDP-43 inclusions | Absent | Absent | Absent | Absent | Absent |
| TMEM106B filaments | Absent | Present | Present | Present | Present |

M, male; F, female; y, years; NA, not applicable; bvFTD, behavioural variant frontotemporal dementia; ALS, amyotrophic lateral sclerosis; FTLD-FUS, frontotemporal lobar degeneration with FUS-immunoreactive inclusions; FTLD-FET, frontotemporal lobar degeneration with FET protein-immunoreactive inclusions; aFTLD-U, atypical frontotemporal lobar degeneration with ubiquitin-positive inclusions; MNP, upper and lower motor neuron pathology; PFC, prefrontal cortex; TC, temporal cortex; MC, motor cortex; SC, spinal cord; BS, brainstem; AD-NC, Alzheimer's disease neuropathologic change; A, Thal phase; B, Braak stage; C, CERAD score.

# Extended Data Table 2 | CryoEM data collection, refinement and validation statistics

| | Individual 1 | | Individual 2 | Individiual 3 | Individual 4 | | | | | | | |
| --- | --- | --- | --- | --- | --- | --- | --- | --- | --- | --- | --- | --- |
| | prefrontal cortex | motor cortex | prefrontal cortex + temporal lobe | prefrontal cortex + temporal lobe | prefrontal cortex | | | | brainstem | | | |
| | (EMD-16999, PDB 8ONS) | (EMD-18236) | (EMD-17022) | (EMD-17021) | (EMD-17020) | (EMD-17109) | (EMD-18240) | (EMD-18242) | (EMD-18227) | (EMD-18226) | (EMD-18243) | (EMD-18241) |
| **Data collection and processing** | | | | | | | | | | | | |
| Identity | TAF15 | TAF15 | TAF15 | TAF15 | TAF15 | amyloid-β 42 | singlet TMEM106B | doublet TMEM106B | TAF15 | amyloid-β 42 | singlet TMEM106B | doublet TMEM106B |
| Voltage (kV) | 300 | 300 | 300 | 300 | 300 | | | | 300 | | | |
| Electron source | XFEG | XFEG | XFEG | XFEG | XFEG | | | | XFEG | | | |
| Detector | Falcon 4 | K3 | K3 | K3 | K3 | | | | Falcon 4i | | | |
| Electron exposure (e–/Å$^2$) | 40 | 40 | 40 | 40 | 40 | | | | 40 | | | |
| Defocus range (μm) | -2.0 to -0.8 | -2.0 to -0.8 | -2.0 to -0.8 | -2.0 to -0.8 | -2.0 to -0.8 | | | | -2.0 to -0.8 | | | |
| Pixel size (Å) | 0.824 | 0.834 | 0.834 | 0.834 | 0.834 | | | | 0.824 | | | |
| Microgrpahs collected | 21,997 | 30,044 | 15,600 | 18,120 | 57,829 | | | | 32,290 | | | |
| Symmetry imposed | C1 | C1 | C1 | C1 | C1 | C1 | C1 | C2 | C1 | C2 | C1 | C2 |
| Initial particle images (no.) | 284,131 | 293,874 | 238,598 | 112,436 | 357,803 | | 144,872 | | 517,075 | | | |
| Final particle images (no.) | 95,582 | 67,089 | 177,709 | 49,849 | 68,613 | 77,484 | 93,600 | 21,641 | 103,820 | 37,034 | 117,061 | 29,006 |
| Helical twist (°) | -2.12408 | -2.18 | -2.15 | -2.12 | -2.15 | -3.06 | -0.43 | -0.43 | -2.15 | -3.06 | -0.44 | -0.42 |
| Helical rise (Å) | 4.78 | 4.81 | 4.78 | 4.71 | 4.78 | 4.78 | 4.81 | 4.82 | 4.80 | 4.82 | 4.84 | 4.80 |
| Map resolution (Å) | 1.97 | 2.27 | 2.14 | 2.63 | 2.67 | 3.05 | 2.93 | 3.56 | 2.54 | 3.70 | 2.39 | 2.97 |
| FSC threshold | 0.143 | 0.143 | 0.143 | 0.143 | 0.143 | 0.143 | 0.143 | 0.143 | 0.143 | 0.143 | 0.143 | 0.143 |
| Map resolution range (Å) | 1.96 to 4.13 | 2.34 to 7.14 | 2.02 to 4.26 | 2.39 to 5.35 | 2.58 to 6.47 | 2.84 to 8.54 | 2.59 to 13.41 | 3.17 to 12.72 | 2.55 to 6.07 | 3.48 to 7.27 | 2.24 to 10.15 | 2.64 to 7.46 |

| **Refinement** | |
| --- | --- |
| Model resolution (Å) | 1.97 |
| FSC threshold | 0.5 |
| Map sharpening $B$ factor (Å$^2$) | -34.1 |
| Model composition | |
| Non-hydrogen atoms | 720 |
| Protein residues | 93 |
| Solvent atoms | 23 |
| $B$ factors (Å$^2$) | |
| Protein | 39.7 |
| Solvent | 46.6 |
| R.m.s. deviations | |
| Bond lengths (Å) | 0.0125 |
| Bond angles (°) | 1.7827 |
| Validation | |
| MolProbity score | 1.22 |
| Clashscore | 1.26 |
| Favoured rotamers | 98.63 |
| Poor rotamers (%) | 0 |
| Ramachandran plot | |
| Favored (%) | 94.51 |
| Allowed (%) | 100 |
| Outliers (%) | 0 |

# Reporting Summary

## Statistics

For all statistical analyses, confirm that the following items are present in the figure legend, table legend, main text, or Methods section.

| n/a | Confirmed | |
|---|---|---|
| ☐ | ☒ | The exact sample size ($n$) for each experimental group/condition, given as a discrete number and unit of measurement |
| ☐ | ☒ | A statement on whether measurements were taken from distinct samples or whether the same sample was measured repeatedly |
| ☒ | ☐ | The statistical test(s) used AND whether they are one- or two-sided<br>*Only common tests should be described solely by name; describe more complex techniques in the Methods section.* |
| ☒ | ☐ | A description of all covariates tested |
| ☒ | ☐ | A description of any assumptions or corrections, such as tests of normality and adjustment for multiple comparisons |
| ☐ | ☒ | A full description of the statistical parameters including central tendency (e.g. means) or other basic estimates (e.g. regression coefficient) AND variation (e.g. standard deviation) or associated estimates of uncertainty (e.g. confidence intervals) |
| ☒ | ☐ | For null hypothesis testing, the test statistic (e.g. $F$, $t$, $r$) with confidence intervals, effect sizes, degrees of freedom and $P$ value noted<br>*Give P values as exact values whenever suitable.* |
| ☐ | ☒ | For Bayesian analysis, information on the choice of priors and Markov chain Monte Carlo settings |
| ☒ | ☐ | For hierarchical and complex designs, identification of the appropriate level for tests and full reporting of outcomes |
| ☒ | ☐ | Estimates of effect sizes (e.g. Cohen's $d$, Pearson's $r$), indicating how they were calculated |

*Our web collection on statistics for biologists contains articles on many of the points above.*

## Software and code

Policy information about availability of computer code

| Data collection | EPU 2.14 |
|---|---|
| Data analysis | DIA-NN 1.8.1, RELION 4.0, CTFFIND 4.1, COOT 0.9.8.2, ISOLDE 1.5, REFMAC 5.8.0387, Servalcat 0.3.0, ChimeraX 1.5, MolProbity 4.5.2. |

For manuscripts utilizing custom algorithms or software that are central to the research but not yet described in published literature, software must be made available to editors and reviewers. We strongly encourage code deposition in a community repository (e.g. GitHub). See the Nature Portfolio guidelines for submitting code & software for further information.

## Data

Policy information about availability of data

All manuscripts must include a data availability statement. This statement should provide the following information, where applicable:

- Accession codes, unique identifiers, or web links for publicly available datasets
- A description of any restrictions on data availability
- For clinical datasets or third party data, please ensure that the statement adheres to our policy

Whole-exome data have been deposited in the National Institute on Ageing Alzheimer's Disease Data Storage Site (NIAGADS), under accession code NG00107. Mass spectrometry data have been deposited to the Proteomics Identifications (PRIDE) database under accession code PXD044821. Cryo-EM datasets have been deposited to the Electron Microscopy Public Image Archive (EMPIAR) under accession codes EMPIAR-11735 (individual 1, prefrontal cortex); EMPIAR-11736 (individual 1, motor cortex); EMPIAR-11737 (individual 2 prefrontal and temporal cortex); EMPIAR-11738 (individual 3, prefrontal and temporal cortex);

EMPIAR-11739 (individual 4, prefrontal cortex); and EMPIAR-11740 (individual 4, brainstem). Cryo-EM maps have been deposited to the Electron Microscopy Data Bank (EMDB) under accession codes EMD-16999 and EMD-18236 (TAF15 filaments from prefrontal cortex and motor cortex, respectively, of individual 1); EMD-17022 (TAF15 filaments from prefrontal and temporal cortex of individual 2); EMD-17021 (TAF15 filaments from prefrontal and temporal cortex of individual 3); EMD-17020 and EMD-18227 (TAF15 filaments from prefrontal cortex and brainstem, respectively, of individual 4); EMD-17109 and EMD-18226 (Aβ42 filaments from prefrontal cortex and brainstem, respectively, of individual 4); EMD-18240 and EMD-18243 (singlet TMEM106B filaments from prefrontal cortex and brainstem, respectively, of individual 4); and EMD-18242 and EMD-18241 (doublet TMEM106B filaments from prefrontal cortex and brainstem, respectively, of individual 4). The atomic model for the TAF15 amyloid filaments has been deposited to the Protein Data Bank (PDB) under accession code 8ONS. The atomic models of singlet and doublet TMEM106B type I filaments are available at the PDB under accession codes 7qvc and 7qvf, respectively. The atomic model of Aβ42 type II filaments is available at the PDB under accession code 7q4m.

## Human research participants

Policy information about studies involving human research participants and Sex and Gender in Research.

| | |
|---|---|
| Reporting on sex and gender | 1 female and 4 males. |
| Population characteristics | See Extended Data Table 1. Between 30 and 79 years-of-age. No neurodegenerative disease associated mutations. Clinical diagnoses of bvFTD and ALS. Neuropathological diagnosis of FTLD-FET. |
| Recruitment | Selected based on availability and neuropathological examination. |
| Ethics oversight | Human tissue samples were from the Brain Library of the Dementia Laboratory at Indiana University School of Medicine and the Queen Square Brain Bank for Neurological Disorders at UCL Queen Square Institute of Neurology. Their use in this study was approved by the ethical review processes at each institution. |

Note that full information on the approval of the study protocol must also be provided in the manuscript.

## Field-specific reporting

Please select the one below that is the best fit for your research. If you are not sure, read the appropriate sections before making your selection.

☒ Life sciences    ☐ Behavioural & social sciences    ☐ Ecological, evolutionary & environmental sciences

For a reference copy of the document with all sections, see nature.com/documents/nr-reporting-summary-flat.pdf

## Life sciences study design

All studies must disclose on these points even when the disclosure is negative.

| | |
|---|---|
| Sample size | Frontotemporal cortex from 4 individuals with FTLD-FET. Samples were chosen based on availability and neuropathological examination. |
| Data exclusions | Pre-established common image classification procedures (Scheres 2012. J. Struc. Biol. 180, 519-530) were employed to select particle images with the highest resolution content in the cryo-EM reconstruction process. Details of the number of selected images are given in Extended Data Table 2. |
| Replication | All attempts at replication were successful. Four independent biological repeats per experiment where representative data are shown, as described in the main text. |
| Randomization | Randomisation was not performed. As the samples were limited by brain availability, randomisation would not have reduced any bias in this study. |
| Blinding | The investigators were not blinded to allocation during experiments and outcome assessment. The perceived risk of detection/performance bias was deemed negligible. |

## Reporting for specific materials, systems and methods

We require information from authors about some types of materials, experimental systems and methods used in many studies. Here, indicate whether each material, system or method listed is relevant to your study. If you are not sure if a list item applies to your research, read the appropriate section before selecting a response.

## Materials & experimental systems

| n/a | Involved in the study |
|---|---|
| ☐ | ☒ Antibodies |
| ☒ | ☐ Eukaryotic cell lines |
| ☒ | ☐ Palaeontology and archaeology |
| ☒ | ☐ Animals and other organisms |
| ☒ | ☐ Clinical data |
| ☒ | ☐ Dual use research of concern |

## Methods

| n/a | Involved in the study |
|---|---|
| ☒ | ☐ ChIP-seq |
| ☒ | ☐ Flow cytometry |
| ☒ | ☐ MRI-based neuroimaging |

## Antibodies

| | |
|---|---|
| Antibodies used | The primary antibodies used were anti-FUS (Proteintech 11570-1-AP), anti-TAF15 (Bethyl IHC-00094), anti-TAF15 (Bethyl A300-308A), anti-EWS (Santa Cruz sc-28327) and anti-transportin-1 (Abcam ab10303). |
| Validation | Antibody validation is presented in the manufacturers' datasheets, as well as in (Neumann et al. 2011. Brain 9, 2595–2609), (Brelstaff et al. 2011. Acta Neuropathol. 5, 591–600), (Neumann et al. 2012. Acta Neuropathol. 5, 705–716) and (Davidson et al. 2013. Neuropathol. Appl. Neurobiol. 2, 157–165). |

