## [Peer Review File · Nature]

Manuscript Title: TAF15 amyloid filaments in frontotemporal lobar degeneration

Reviewer Comments & Author Rebuttals

Reviewer Reports on the Initial Version:

Referees' comments:

Referee #1 (Remarks to the Author):

A unifying theme for neurodegenerative disorders is the abnormal accumulation of protein aggregates in the central nervous systems of affected individuals. Over the last ~20 years, the isolation of the proteinaceous building blocks of these pathological deposits has provided tremendous insight into disease mechanisms. Genetics discoveries have often gone hand in glove with the biochemical discoveries and indeed mutations in several of the genes encoding the aggregation-prone proteins are causative for rare familial forms of the disease (for example, APP mutations cause early onset AD, alpha-synuclein mutations cause early onset PD, tau mutations cause frontotemporal dementia, and TDP-43 mutations cause ALS). All of this is to say that identifying a new aggregated protein in a neurodegenerative disease is of great interest and will likely be highly informative and impactful.

For frontotemporal lobar degeneration (FTLD), cases can be divided into three major types, based on pathology: 1) FTLD-Tau, which is characterized by tau inclusions; 2) FTLD-TDP-43, which is characterized by TDP-43 inclusions; and a rarer subtype, FTLD-FUS, which is characterized by FUS inclusions.

This manuscript by Tetter et al is a short paper essentially presenting one new finding: presenting TAF15 as the disease protein FTLD-FUS cases (contrary to it being FUS as expected). The authors isolate amyloid material from the brains of 4 FTLD-FUS patients and apply Cryo-EM approaches to obtain models of the filamentous material, revealing it to be TAF15. The data presented in Fig. 1 is not novel. It has been known that TAF15 pathology is a component of FTLD-FUS cases and is known as FTLD-FET (for FUS, EWSR1, TAF15). Those three RNA-binding proteins are very similar to each other. However, the Cryo-EM studies (not my expertise) and other analyses show that it is only TAF15 but not EWSR1 or FUS (or any other amyloid) in the deposits – this is VERY exciting and novel and of great interest. It really changes how the field will think about this disease and will launch efforts to study TAF15 and perhaps target it for therapy. This finding might help to explain the different observations between FTLD-FUS (not caused by FUS mutations and FUS, EWSR1, TAF15, and transportin1 are present in the inclusions) and ALS-FUS (caused by FUS mutations and only FUS in the inclusions).

Even though this paper is very short with really only 1 figure of novel data (Fig. 2 and 3 could probably be combined into one figure and Fig. 1 confirms previous observations), in my opinion this discovery is extremely important and will be of great interest to the readers of Nature and I strongly suspect that this will motivate launching of drug discovery programs targeting TAF15 with ASOs as a therapy for this rare form of FTLD. I have some comments and suggestions for the authors to consider.

1) ALS-FUS cases are rare but there are patient samples available. It would be informative if the authors could apply similar isolation procedures on these samples (preferably ones with different mutations and aggressive forms and slower forms) and test if it is TAF15 amyloids or only FUS amyloids in these.

2) The authors isolate TAF15 amyloids from four patients. Is it feasible to add additional FTLD-FET

samples to explore how robust this new discovery is?

3) The authors hypothesize that TAF15 amyloids might sequester FUS, EWSR1, and transportin1 proteins into pathological conclusions. Could the authors test this hypothesis in cell culture experiments (i.e., introducing TAF15 amyloids into cells and testing if this causes mislocalization of the other endogenous proteins?

Referee #2 (Remarks to the Author):

Tetter et al. report the structure of TAF15 fibrils that were purified from FTDL-FUS patient brains. However, instead of FUS protein forming the fibrils, the protein TAF15 was identified based on the obtained structure. For this, the authors have employed Sarkosyl-extraction in combination with cryo-electron microscopy to resolve the high-resolution structures of inclusion body fibrils derived from the brain tissue of four patients diagnosed with FTLD-FUS. Fibrils in all cases had the same fold. The cryo-EM results are of excellent quality, and the density of the TAF15 LCD is in excellent agreement with the sequence.

In lines 89-92, the authors state that other fibrils observed are TMEM106B, based on their diameter of 12 or 26 nm and their surface texture. This is a rather weak argument for the identification of the fibril proteins. Could these be a different fibril form of TAF15? Or could these fibrils be the FUS that was initially expected? (In line 177-179, the authors even speculate about this possibility: FUS may form filaments that are not captured by our extraction method, possibly due to differences in stability or solubility.).

The authors should provide a high-resolution 3D reconstruction of these fibrils to clarify if this is indeed TMEM106B and not a fibril of some other protein.

The authors could also do some biochemical characterization of FUS from the soluble fraction to determine if it is present in a fibrillar or non-fibrillar form.

The authors mentioned that TMEM106B filaments were previously reported to accumulate in healthy brains and therefore were not analyzed any further. What about TAF15, is there any evidence of whether it also accumulates in healthy brains or not?

The authors propose in lines 220/221 to refer to FTLD-FUS cases as FTLD-TAF. This may be a premature statement, as the clinical evidence for TAF-15 fibrils as disease causing is not yet clear. Such a renaming would contribute to the confusion in the field at this stage.

Minor:

The information is missing, how many micrographs were being processed for each sample.

The extracted samples are worth sending to mass spectrometry to gain additional sequence information of those proteins.

Referee #3 (Remarks to the Author):

The manuscript by Tetter et al. describes the structure of amyloid fibrils determined by cryoEM of fibrils extracted from four individuals with frontotemporal lobar degeneration (FTLD). It was widely accepted that rare cases of FTLD involve fibril formation of the protein FUS. Using the sarkosyl

extraction method developed by Scheres and colleagues, Tetter et al. unexpectedly find that the fibrils of the four rare cases of FTLD involve amyloid fibrils formed from the FUS homologue called TAF15. The results accord with previous reports of TAF 15 in FTLD-FUS and add a new structure of these fibrils.

The results present high resolution structures of the TAF15 amyloid fibrils and technically are well done and are well described.

However, I did not feel that the work presents a finding of the novelty and conceptual importance I would expect to see in a manuscript in Nature. The fibril fold is new and adds to the wealth of fibril folds of brain amyloid deposits associated with neurodegeneration that the group of Scheres and Goedert at the LMB and elsewhere have now solved. There are bound waters and bound non-protein density that has been widely seen in such fibrils *ex vivo*. That the same fibril fold is found in four individuals is interesting, but not conceptually new, in that, at least for neurodegeneration, different fibril folds appear to be found in individuals with the same disorder. Indeed, this study adds to the growing wealth of data on that topic.

The study will be of interest to those focused on FTLD-FUS and amyloid structure. I think it would be suited to Nature Communications or other such journal wherein it will have a good reception to those specialised in amyloid structure and FTLD-FUS fields.

Author Rebuttals to Initial Comments:

We thank the Referees for their constructive and insightful comments, which we feel have improved the manuscript and substantially enhanced the impact of the work. Please find our point-by-point responses to the individual comments below, written in blue text.

Referee #1 (Remarks to the Author):

A unifying theme for neurodegenerative disorders is the abnormal accumulation of protein aggregates in the central nervous systems of affected individuals. Over the last ~20 years, the isolation of the proteinaceous building blocks of these pathological deposits has provided tremendous insight into disease mechanisms. Genetics discoveries have often gone hand in glove with the biochemical discoveries and indeed mutations in several of the genes encoding the aggregation-prone proteins are causative for rare familial forms of the disease (for example, APP mutations cause early onset AD, alpha-synuclein mutations cause early onset PD, tau mutations cause frontotemporal dementia, and TDP-43 mutations cause ALS). All of this is to say that identifying a new aggregated protein in a neurodegenerative disease is of great interest and will likely be highly informative and impactful.

For frontotemporal lobar degeneration (FTLD), cases can be divided into three major types, based on pathology: 1) FTLD-Tau, which is characterized by tau inclusions; 2) FTLD-TDP-43, which is characterized by TDP-43 inclusions; and a rarer subtype, FTLD-FUS, which is characterized by FUS inclusions.

This manuscript by Tetter et al is a short paper essentially presenting one new finding: presenting TAF15 as the disease protein FTLD-FUS cases (contrary to it being FUS as expected). The authors isolate amyloid material from the brains of 4 FTLD-FUS patients and apply Cryo-EM approaches to obtain models of the filamentous material, revealing it to be TAF15. The data presented in Fig. 1 is not novel. It has been known that TAF15 pathology is a component of FTLD-FUS cases and is known as FTLD-FET (for FUS, EWSR1, TAF15).

We did not intend to claim that our immunohistochemistry results in Figure 1 are novel. We have cited the five publications that previously reported FET protein and transportin 1 immunoreactivity of the inclusions of FTLD-FUS and that propose the use of FTLD-FET (references 22–26 in the revised manuscript), both in the introduction and alongside our results relating to Figure 1. To further emphasise this, we have replaced the word 'revealed' with 'confirmed the presence of' when reporting these results (line 80). However, we feel that it is

important to show these results to confirm that the cases we use for cryo-EM are bona fide FTLD-FET.

Those three RNA-binding proteins are very similar to each other. However, the Cryo-EM studies (not my expertise) and other analyses show that it is only TAF15 but not EWSR1 or FUS (or any other amyloid) in the deposits – this is VERY exciting and novel and of great interest. It really changes how the field will think about this disease and will launch efforts to study TAF15 and perhaps target it for therapy. This finding might help to explain the different observations between FTLD-FUS (not caused by FUS mutations and FUS, EWSR1, TAF15, and transportin1 are present in the inclusions) and ALS-FUS (caused by FUS mutations and only FUS in the inclusions).

Even though this paper is very short with really only 1 figure of novel data (Fig. 2 and 3 could probably be combined into one figure and Fig. 1 confirms previous observations), in my opinion this discovery is extremely important and will be of great interest to the readers of Nature and I strongly suspect that this will motivate launching of drug discovery programs targeting TAF15 with ASOs as a therapy for this rare form of FTLD.

I have some comments and suggestions for the authors to consider.

1) ALS-FUS cases are rare but there are patient samples available. It would be informative if the authors could apply similar isolation procedures on these samples (preferably ones with different mutations and aggressive forms and slower forms) and test if it is TAF15 amyloids or only FUS amyloids in these.

As discussed in the manuscript (line 275), the motor neuron inclusions in cases of ALS caused by mutations in *FUS* do not contain TAF15, EWS or transportin 1 (references 23 and 25 in the revised manuscript). Therefore, it is unlikely that TAF15 forms amyloid filaments in these cases, but possible that these inclusions may contain FUS amyloid filaments. We have added a sentence to this effect to the manuscript (line 276). As discussed in the manuscript (line 278), this supports the hypothesis that the disease mechanisms of ALS caused by *FUS* mutations are fundamentally different to those of FTLD-FET. Additional evidence for distinct disease mechanisms comes from the observation that FUS is hypomethylated in FTLD-FET, but not in ALS caused by *FUS* mutations (reference 51 in the revised manuscript), which we have now cited in the Discussion (line 280). Based on our previous experience, obtaining samples and

performing cryo-EM would take in excess of one year. We have added a sentence to the Discussion (line 281) to propose that future studies should investigate the presence and structures of amyloid filaments from cases of this rare form of familial ALS to further test this hypothesis.

In contrast, motor neuron inclusions that are immunoreactive against the FET proteins and transportin 1 have been reported in cases of sporadic ALS without *FUS* mutations (references 47 and 48 in the revised manuscript). Furthermore, individuals with FTLD-FET often have some degree of FET protein- and transportin 1-immunoreactive inclusions in upper and lower motor neurons (references 12, 15 and 23 in the revised manuscript). These studies led us to speculate that TAF15 might form amyloid filaments in some cases of ALS and that these cases might be part of a disease spectrum with FTLD-FET, in analogy to ALS-TDP and FTLD-TDP (references 1, 2 and 5 in the revised manuscript).

To test this hypothesis, we carried out a detailed evaluation of upper and lower motor neuron pathology in the four individuals. We found that individuals 1 and 4, but not 2 and 3, showed upper and lower motor neuron pathology. Individuals 1 and 4 showed the loss of upper and lower motor neurons. Luxol fast blue staining of myelin also showed extensive corticospinal tract loss for individual 1. Corticospinal tract loss could not be assessed for individual 4 due to a lack of available spinal cord. We have added these new findings to the Methods section (line 319 of the revised manuscript) and to a new Extended Data Figure 8. We had also noted in the manuscript that individual 4 indeed presented with probable ALS in addition to FTD.

We next performed immunohistochemistry for the FET proteins and for transportin 1 on motor system tissues (spinal cord, motor cortex and brainstem) from the four individuals. *FUS*-, TAF15- and transportin 1-immunoreactive inclusions were readily observed in upper and lower motor neurons for individuals 1 and 4, but were sparse or absent for individuals 2 and 3. We have added these new results to a new Figure 4 and to the new Extended Data Figure 8.

We proceeded to determine the cryo-EM structures of amyloid filaments extracted from the motor cortex of individual 1 and from the brainstem of individual 4. For both individuals, we found abundant TAF15 filaments with the same fold as those in the prefrontal and temporal cortices of the four individuals. We did not find filaments of *FUS*. For individual 4, we also found TMEM106B and A β 42 filaments with the same folds as in the prefrontal cortex, in

addition to TAF15 filaments, consistent with the older age of this individual and the presence of sparse neuritic plaques. No TMEM106B or A β 42 filaments were found for individual 1, mirroring our results from the prefrontal cortex and consistent with the younger age of this individual and absence of neuritic plaques. We have added these results to the new Figure 4 and to a new Extended Data Figure 9.

Together, these results suggest that the formation of TAF15 amyloid filaments can be associated with upper and lower motor neuron pathology and may underlie a disease spectrum of FTLD and ALS, in analogy to TDP-43. We have added new sections to the Results (line 184) and Discussion (line 207) to present and discuss these new results that link TAF15 amyloid filaments to ALS, in addition to FTLD.

2) The authors isolate TAF15 amyloids from four patients. Is it feasible to add additional FTLD-FET samples to explore how robust this new discovery is?

We do not have access to flash frozen samples from additional individuals. Based on our previous experience, we estimate that sourcing samples and determining cryo-EM structures would take in excess of one year to complete. However, as detailed in our response to point 1, we have now performed cryo-EM on filaments extracted from the motor system of two of the individuals and find TAF15 filaments with the same fold as we observed in the prefrontal and temporal cortices of the four individuals. In addition, this study represents the largest number of individuals with a given neurodegenerative condition used for cryo-EM to date. Previous studies used a maximum of three individuals for a given condition (e.g. for tau, Shi et al. 2021. Nature 598, 359–363; for α -synuclein, Yang et al. 2022. Nature 610, 791–795; for TDP-43, Arseni et al. 2023. Nature doi.org/10.1038/s41586-023-06405-w; for A β 42, Yang et al. 2022. Science 375, 167-172; and for TMEM106B, Schweighauser et al. 2022. Nature 605, 310–314).

3) The authors hypothesize that TAF15 amyloids might sequester FUS, EWSR1, and transportin1 proteins into pathological conclusions. Could the authors test this hypothesis in cell culture experiments (i.e., introducing TAF15 amyloids into cells and testing if this causes mislocalization of the other endogenous proteins)?

We agree that this is an important future research direction resulting from our findings. However, this represents a major study that may take years to complete. This is because the proposed study requires a cell culture model that recapitulates the TAF15 amyloid filament

fold described here, which currently does not exist. Existing cell culture models of amyloid formation of other proteins, such as tau, do not recapitulate disease-associated filament folds (e.g. Tarutani et al. 2023. FEBS Open Bio. 13, 1394–1404). As such, this work is beyond the scope of this study. We have added a sentence to the Discussion (line 246) to acknowledge that this is an important future research direction and that cell culture models that recapitulate disease-associated filament folds are required.

Referee #2 (Remarks to the Author):

Tetter et al. report the structure of TAF15 fibrils that were purified from FTDL-FUS patient brains. However, instead of FUS protein forming the fibrils, the protein TAF15 was identified based on the obtained structure. For this, the authors have employed Sarkosyl-extraction in combination with cryo-electron microscopy to resolve the high-resolution structures of inclusion body fibrils derived from the brain tissue of four patients diagnosed with FTLD-FUS. Fibrils in all cases had the same fold. The cryo-EM results are of excellent quality, and the density of the TAF15 LCD is in excellent agreement with the sequence.

In lines 89-92, the authors state that other fibrils observed are TMEM106B, based on their diameter of 12 or 26 nm and their surface texture. This is a rather weak argument for the identification of the fibril proteins. Could these be a different fibril form of TAF15? Or could these fibrils be the FUS that was initially expected? (In line 177-179, the authors even speculate about this possibility: FUS may form filaments that are not captured by our extraction method, possibly due to differences in stability or solubility.). The authors should provide a high-resolution 3D reconstruction of these fibrils to clarify if this is indeed TMEM106B and not a fibril of some other protein.

We have now determined the cryo-EM structures of the single and double TMEM106B filaments from the prefrontal cortex of individual 4 to resolutions of 2.9 and 3.6 Å, respectively. In addition, related to our response to point 1 from Referee 1, we have also determined the cryo-EM structures of the single and double TMEM106B filaments from the brainstem of this individual to resolutions of 2.4 and 2.8 Å, respectively. These represent the highest-resolution structures of TMEM106B filaments determined to date. These structures confirm that this filament population is formed of TMEM106B and reveal that they have the type I TMEM106B filament fold, as described in (Schweighauser et al. 2022. Nature 605, 310–314). We have

added these new results to the revised manuscript (lines 102 and 200) and to Extended Data Figures 2 and 9.

The authors could also do some biochemical characterization of FUS from the soluble fraction to determine if it is present in a fibrillar or non-fibrillar form.

We have now added negative-stain electron micrographs of the soluble and insoluble fractions from the prefrontal cortex of individual 1. We did not find filaments in the soluble fraction, whereas filaments were detected in the insoluble fraction as expected. This indicates that FUS is likely to be present in a non-filamentous form in the soluble fraction. We have added these new results to the revised manuscript (line 88) and to Extended Data Figure 1.

The authors mentioned that TMEM106B filaments were previously reported to accumulate in healthy brains and therefore were not analyzed any further. What about TAF15, is there any evidence of whether it also accumulates in healthy brains or not?

There is no evidence that TAF15 filaments accumulate in healthy brains. TAF15-immunoreactive neuronal cytoplasmic inclusions are unique to FTLN-FET/ sporadic ALS-FET and have never been observed in other neurodegenerative conditions or in neurologically normal brains (references 23–27 in the revised manuscript). Furthermore, we did not find TAF15 filaments in other neurodegenerative conditions or in neurologically normal brains in our previous work on TMEM106B filaments, where we took a similar unbiased approach to identifying amyloid filaments (reference 38 in the revised manuscript). We have added a sentence to make this explicit in the revised manuscript (line 212).

The authors propose in lines 220/221 to refer to FTLN-FUS cases as FTLN-TAF. This may be a premature statement, as the clinical evidence for TAF-15 fibrils as disease causing is not yet clear. Such a renaming would contribute to the confusion in the field at this stage.

We followed the recommendations of the *'Nomenclature for neuropathologic subtypes of frontotemporal lobar degeneration: consensus recommendation'* (reference 52 in the revised manuscript), which states that naming, "should be designated by the protein abnormality that is presumed to be pathogenic or most characteristic of the condition (i.e. FTLN-protein)," and that, "when a new entity is discovered or when the molecular identity of the major pathological factor in an existing group is clarified, the appropriate term will be FTLN- pathological molecule".

However, it is not our intention to cause confusion. We have, therefore, amended this sentence (line 297) to place the emphasis of the inappropriateness of the term FTLD-FUS, to recommend the use of FTLD-FET, and to only tentatively put forward the term FTLD-TAF. Additionally, we have also decided to use “FTLD-FET” throughout the manuscript after the first mention of that terminology, in order to not propagate the misleading “FTLD-FUS” nomenclature.

Minor:

The information is missing, how many micrographs were being processed for each sample. We have now added this information to Extended Data Table 2.

The extracted samples are worth sending to mass spectrometry to gain additional sequence information of those proteins.

We have now performed mass spectrometry on extracted samples from individuals 1 and 4, as well as from a neurologically normal control individual. We found peptides from the region that forms the TMEM106B filament core in samples from individual 4 and an aged neurologically normal individual, but not from individual 1. We also found peptides corresponding to A β 42 in extracts from individual 4, but not from individual 1 or the neurologically normal individual. Finally, we found that while peptides mapping to FUS and TAF15 were detected for all individuals, only those mapping to the region that forms of core of TAF15 filaments could distinguish between the individuals with FTLD-FET and the neurologically normal individual. These new results support our cryo-EM results. We have added these new results to the revised manuscript (lines 104, 120 and 134) and to a new Extended Data Figure 3.

Referee #3 (Remarks to the Author):

The manuscript by Tetter et al. describes the structure of amyloid fibrils determined by cryoEM of fibrils extracted from four individuals with frontotemporal lobar degeneration (FTLD). It was widely accepted that rare cases of FTLD involve fibril formation of the protein FUS. Using the sarkosyl extraction method developed by Scheres and colleagues, Tetter et al. unexpectedly find that the fibrils of the four rare cases of FTLD involve amyloid fibrils formed from the FUS

homologue called TAF15. The results accord with previous reports of TAF 15 in FTL-D-FUS and add a new structure of these fibrils.

We would like to reiterate that TAF15 was not known to form amyloid filaments in neurodegenerative disease prior to this study. Moreover, no structures of TAF15 filaments existed prior to this study (either *in vitro* or *ex vivo*). Existing reports of TAF15-immunoreactivity of the inclusions in FTL-D-FET (references 22–26 in the revised manuscript) is consistent with our finding of TAF15 filaments. However, inclusion immunoreactivity is not sufficient to make conclusions about filament formation, because many non-filamentous proteins are present in such inclusions (e.g. reference 45 in the revised manuscript). It was only possible to make the discovery that TAF15 forms amyloid filaments in neurodegenerative disease using cryo-EM.

The results present high resolution structures of the TAF15 amyloid fibrils and technically are well done and are well described.

However, I did not feel that the work presents a finding of the novelty and conceptual importance I would expect to see in a manuscript in Nature. The fibril fold is new and adds to the wealth of fibril folds of brain amyloid deposits associated with neurodegeneration that the group of Scheres and Goedert at the LMB and elsewhere have now solved. There are bound waters and bound non-protein density that has been widely seen in such fibrils *ex vivo*. That the same fibril fold is found in four individuals is interesting, but not conceptually new, in that, at least for neurodegeneration, different fibril folds appear to be found in individuals with the same disorder. Indeed, this study adds to the growing wealth of data on that topic.

The study will be of interest to those focused on FTL-D-FUS and amyloid structure. I think it would be suited to Nature Communications or other such journal wherein it will have a good reception to those specialised in amyloid structure and FTL-D-FUS fields.

We feel that the interpretation of these results as *just another cryo-EM structure of amyloid filaments* indicates that we did not sufficiently highlight the fundamental importance of this study. Prior to this work, TAF15 was not known to form amyloid filaments in neurodegenerative disease and no structures existed. This work represents a rare finding of a new member of the small group of proteins known to form neurodegenerative disease-associated amyloid filaments, alongside proteins such as tau (Lee et al. 1991. Science 251, 675–678), α -synuclein (Spillantini et al. 1997. Nature 388, 839–840) and TDP-43 (Neumann

et al. 2006. *Science* 314, 130–133). The discovery of these proteinopathies catalysed significant efforts to investigate their genetics, neuropathology, biochemistry, cellular biology and pharmacology (with 9,561 PubMed citations since 2020 for "tau protein," 5,531 for "alpha-synuclein" and 1,575 for "TDP-43"). This effort is now translating into promising diagnostic and therapeutic strategies. We predict that our fundamental finding that TAF15 amyloid filaments are associated with neurodegenerative disease will have a similar transformation effect (as of today, there are only 86 PubMed citations concerning "TAF15" since 2020). For this reason, we believe that a journal of such broad readership as *Nature* is appropriate. To emphasise this fundamental discovery over secondary insights into structural details of the TAF15 filament fold, we have revised the Summary (line 29), Discussion (line 214) and Conclusion (line 294) to add further emphasis.

Reviewer Reports on the First Revision:

Referees' comments:

Referee #1 (Remarks to the Author):

The authors have addressed my comments and suggestions and have even solved cryo-EM structures for TAF15 filaments from two additional patients (brainstem and motor cortex), further demonstrating the presence of TAF15 amyloid filaments in FTD-FET cases.

It is of course impossible to predict if TAF15 filaments are directly causing neurodegeneration, but there is strong evidence from many other disease proteins (e.g., PrP, alpha-synuclein, Abeta, tau, mutant Htt, mutant SOD1, TDP-43, FUS, etc.) that identifying the disease proteins that form amyloid filaments will provide insight into disease and could represent novel therapeutic targets.

I strongly suspect that this paper will launch new studies from multiple teams to intensely focus on how TAF15 might contribute to disease and how to target TAF15 filaments. I therefore strongly support publication of this novel work in Nature.

Referee #2 (Remarks to the Author):

Tetter et al. have in response to earlier reviewer comments expanded the previous manuscript, which now includes several additional cryo-EM structures of TAF15, TMEM106B, and Abeta. The structures are of excellent quality and resolution. They prove that the obtained structures are TAF15 and that other fibrils were correctly distinguished from the TAF15 filaments as being TMEM106B or Abeta, as the authors already previously had claimed.

The authors have also addressed all previous reviewer comments to my fullest satisfaction. The technical quality of this work is outstanding.

Referee #3 (Remarks to the Author):

In their revisions, the authors have addressed many of the points of the referees, including new structures of TMEM106B fibrils, new structures of TAF fibrils from different regions and MS data to validate the proteins present in soluble and insoluble fractions. Editorial changes and new Figs and Supp Figs have enhanced the story and made the novelty of their finding of TAF15 amyloid. As ever, the cryoEM data are of excellent quality. The authors are much clearer about the importance of their new finding that TAF self-assembles into amyloid in disease and that this will stimulate new understanding into how to diagnose and potentially treat the spectrum of disorders annotated as FLTD-FUS (now FTLD-FET). This is clearly the novel aspect of the work, which could only have been made using the powers of cryoEM to solve the amyloid structures.